# Catechin versus MoS_2_ Nanoflakes Functionalized with Catechin: Improving the Sperm Fertilizing Ability—An In Vitro Study in a Swine Model

**DOI:** 10.3390/ijms24054788

**Published:** 2023-03-01

**Authors:** Costanza Cimini, Marina Ramal-Sanchez, Angela Taraschi, Flavio Della Pelle, Annalisa Scroccarello, Ramses Belda-Perez, Luca Valbonetti, Paola Lanuti, Marco Marchisio, Mario D’Atri, Claudio Ortolani, Stefano Papa, Giulia Capacchietti, Nicola Bernabò, Dario Compagnone, Barbara Barboni

**Affiliations:** 1Department of Biosciences and Technology for Food, Agriculture and Environment, University of Teramo, 64100 Teramo, Italy; 2Department of Innovative Technologies in Medicine and Dentistry, University of Chieti-Pescara, 66100 Chieti, Italy; 3Institute of Biochemistry and Cell Biology (CNRIBBC/EMMA/Infrafrontier/IMPC), National Research Council, 00015 Rome, Italy; 4Department of Medicine and Aging Science, University “G. d’Annunzio” of Chieti-Pescara, 66100 Chieti, Italy; 5Centre on Aging Sciences and Translational Medicine (Ce.S.I-MeT), University “G. d’Annunzio” of Chieti-Pescara, 66100 Chieti, Italy; 6Department of Biomolecular Sciences, University of Urbino “Carlo Bo”, 61029 Urbino, Italy; 7Sharp Solutions Software di D’Atri Mario, Via Udine, 2, Buttrio, 33042 Udine, Italy

**Keywords:** molybdenum disulfide, catechins, spermatozoa, sperm capacitation, in vitro fertilization

## Abstract

Nowadays, the adoption of In Vitro Fertilization (IVF) techniques is undergoing an impressive increase. In light of this, one of the most promising strategies is the novel use of non-physiological materials and naturally derived compounds for advanced sperm preparation methods. Here, sperm cells were exposed during capacitation to MoS_2_/Catechin nanoflakes and catechin (CT), a flavonoid with antioxidant properties, at concentrations of 10, 1, 0.1 ppm. The results showed no significant differences in terms of sperm membrane modifications or biochemical pathways among the groups, allowing the hypothesis that MoS_2_/CT nanoflakes do not induce any negative effect on the parameters evaluated related to sperm capacitation. Moreover, the addition of CT alone at a specific concentration (0.1 ppm) increased the spermatozoa fertilizing ability in an IVF assay by increasing the number of fertilized oocytes with respect to the control group. Our findings open interesting new perspectives regarding the use of catechins and new materials obtained using natural or bio compounds, which could be used to implement the current strategies for sperm capacitation.

## 1. Introduction

In vitro fertilization (IVF) is one of the most used assisted reproductive techniques, aimed at overcoming fertility problems, either in zootechnics or for human purposes. In this process, an egg is combined with spermatozoa in vitro, after the acquisition of their fertilizing potential in a process commonly known as capacitation.

Recently, the use of non-physiological materials is gaining ground in the reproductive field as a support for the implementation of IVF techniques. For instance, previous studies demonstrated a significant improvement in the fertility outcomes when sperm cells were exposed to graphene oxide (GO) during capacitation in different animal models such as swine, bovine and mouse. This effect could probably be ascribed to the extraction of cholesterol from the sperm membrane thus inducing an intense lipid membrane remodeling [1,2,3]. Among non-physiological materials and naturally derived compounds, molybdenum disulfide (MoS_2_) and catechin stand out as interesting candidates to characterize the effects of their exposure on reproductive function.

MoS_2_ is a member of the transition metal dichalcogenides (TMDs) family, which are layered materials with a structure consisting of overlapped layers held together by Van der Waals forces. Each sheet possesses a wafer-like structure with a central hexagonal layer of metal atoms sandwiched in a double layer of chalcogen atoms [4]. MoS_2_ is characterized by both a peculiar nanostructure and chemistry that has commenced to be employed in different fields, including catalysis, electrochemistry and an ever increasing use in biological/biomedical applications [5,6,7]. TMDs’ features are strictly dependent on the nanoscale reduction strategy and despite the impressive advancements achieved during the last years, their use for biological/biomedical applications is often limited by their scarce dispersibility in water and/or the need for toxic or pollutant chemicals for their synthesis. It is noteworthy that MoS_2_ is a 2D graphene-like material and, although the potential toxicity of TMDs materials has been studied in embryonated eggs [8] and on human lung carcinoma epithelial cells [9], to the best of our knowledge there are no studies evaluating the potential effects of TMDs on sperm capacitation.

On the other hand, catechin (CT) is a flavonoid characterized by a high antioxidant capacity [10], and it has previously been demonstrated that CT supplementation to sperm storage may have a beneficial effect on sperm motility [11]. Interestingly, CT has proved to be able to assist the stabilization and synthesis of various nanomaterials, remaining firmly anchored on their surface and acting as a functional and stabilizing agent [12,13]. In the nanomaterials domain, the dispersion or exfoliation route represents a challenge to modulate the affinity and dispersibility of the materials in different media, defining the final material features in terms of structure, dimension and solubility and thus conferring additional functionalities [4,14,15,16].

In the present study, we aimed to study the effects of nanoflakes of MoS_2_ functionalized with catechins and catechins on swine spermatozoa functional parameters during capacitation. The term capacitation encompasses a necessary series of events occurring naturally in vivo and by which spermatozoa undergo a functional modification, ultimately acquiring their fertilizing ability. While in vivo, sperm cells are free to migrate through the uterus, bind to the oviductal epithelium and encounter the multiple endocrine stimuli prior to the meeting with the oocyte [17,18,19]. In order to improve the current strategies for sperm capacitation, different concentrations of MoS_2_/CT and CT were evaluated (10, 1, and 0.1 ppm). A multiple-step approach was adopted to evaluate the potential effects of this interaction in terms of: (a) acrosome damage; (b) membrane disorder; (c) biochemical patterns (PKA activity and tyrosine phosphorylation patterns); (d) intracellular calcium concentration; and (e) mitochondrial activity. As a functional test, finally an IVF assay was performed to assess the sperms’ fertilizing ability.

## 2. Results and Discussion

Here, the potential effects of MoS_2_/CT and CT addition during spermatozoa capacitation were analyzed using a swine in vitro model. The swine model has acquired enormous importance for biomedical research and represents an optimal animal model for the study of human reproductive events [20,21,22].

### 2.1. Preparation and Characterization of Water-Phase Exfoliated MoS_2_/CT Nanoflakes

As reported in Figure 1, the bulk-MoS_2_ sonochemical liquid phase exfoliation assisted by catechin (conducted according to the Section 3 Materials and Methods) allowed a stable colloidal dispersion of MoS_2_/CT nanoflakes to be obtained, which was then used for further experiments. As expected, in the bulk form this TMD possesses crystalline structures characterized by micrometric sides and thickness (Figure 1B); in this conformation, MoS_2_ is not dispersible in water. Figure 1C,D show the SEM micrograph of the MoS_2_ after exfoliation assisted by catechin (MoS_2_/CT). The CT-assisted exfoliated MoS_2_ flakes resulted in being significantly smaller when compared to the bulk MoS_2_, proving a noticeable exfoliation success. In this conformation, the MoS_2_/CT is a colloid, stable for more than 1 year (Figure 1C, right). In the SEM magnification of Figure 1D, the MoS_2_/CT flakes obtained are visible and are characterized by nano sides. The catechin effectiveness in the exfoliation of MoS_2_ was proved in our previous work [12], where the exfoliation strategy was proposed, optimized, and the nanoflakes obtained were fully characterized. An average size value of about 153 ± 2 nm was obtained via dynamic light scattering for the MoS_2_/CT flakes. It is noteworthy that the same study highlighted an interesting residual antioxidant potential for MoS_2_/CT, which was attributed to the MoS_2_ surface modification influenced by the catechin, able to bring charges and redox-active moieties.

Several exfoliation strategies have been proposed for layered materials. Nevertheless, liquid-phase exfoliation (LPE) has become an affordable and sustainable large-scale strategy to produce water-dispersed 2D nanomaterials. Two-dimensional nano colloids have been obtained in water using LPE by employing different surfactants, polymeric structures and different amphiphilic compounds, which are able to interact with the dispersed/exfoliated nanostructures mainly via non-covalent interactions [23,24,25]. Recently, our group has demonstrated how naturally-derived polyphenols can assist the graphene and group VI TMDs’ exfoliation in water, acting as stabilizing agents and conferring at the same time additional features, partially preserving their antioxidant moieties [12,13].

Catechin is a flavonoid with an amphiphilic structure that is naturally present in different foods, characterized by a high antioxidant capacity and thus associated with several potential biological functions and health benefits [10]. Thanks to its amphiphilic structure, catechin can act as a stabilizing agent for nanomaterials’ production and stabilization in water. Catechin has demonstrated an active role in the formation, stabilization and functionalization of metal nanoparticles, graphene, and nanocomposites. Moreover, the catechin adhesion on the formed nanomaterials apports additional electrochemical and antimicrobial features [12,15,26,27], proving to be a useful naturally-derived functional and stabilizing agent.

The sonochemical preparation in water of the MoS_2_/CT can be resumed as follow: (i) the ultrasound energy allows the layers and dimension number/size reductions of the crystalline bulk-MoS_2_; (ii) the stabilization/functionalization of the formed MoS_2_ nanoflakes is guaranteed by the catechins’ ability to remain attached to the surface of the produce MoS_2_ flakes acting as a stabilizing agent. In this case, the catechin carbon skeleton allows π–π interactions with neo-produced MoS_2_ flakes, while the CT hydroxyl groups interact with water via hydrogen bonds allowing the nanoflakes to remain dispersed, ensuring also charge repulsion among flakes and avoiding layer re-stacking. The ability of the compounds containing catechol groups, to allow π–π stacking with other π systems has been already proved by Saiz-Poseu and colleagues (2019) [28] and by Silveri and colleagues (2021) [13] for graphene; on the other hand, electrochemical studies supported the hypothesis that the catechol of catechin is attached to the MoS_2_ nanoflakes but has antioxidant moieties free to react [12].

The water-dispersible nature of MoS_2_/CT nanoflakes allowed easy nanomaterial handling via differential centrifugation. This important feature enables (i) purifying the exfoliated nanoflakes, (ii) removing the excess supernatant, and (iii) resuspending the MoS_2_/CT in the spermatozoa capacitating medium at the desired concentration. This is a challenge for TMDs exfoliated in organic solvents, which are not water dispersible. However, different exfoliation strategies in water imply the use of highly toxic or pollutant chemicals [29]. Unlike the extensively studied graphene oxide, which is naturally dispersible in water, probably the lack of biological studies on spermatozoa with TMDs is attributable to (i) the difficult manipulation of TMDs in aqueous media, (ii) the low-optimal TMDs’ exfoliation (studies are often conducted on just roughly dispersed TMDs), (iii) high toxicity materials produced via conventional approaches.

### 2.2. MoS_2_/CT and CT Supplementation Does Not Affect the Acrosome Integrity

The first step to examine the potential impact of MoS_2_/CT and CT exposure on spermatozoa was the monitoring of the acrosome damage. Because spermatozoa that have lost their acrosome are unable to fertilize an egg, depending on the species, the acrosome reaction (AR) is one of the most important fertilization mechanisms [30].

Acrosome integrity was evaluated at 0 and 1.5 h after in vitro capacitation. As evidenced in Figure 2, after 1.5 h of capacitation in the presence of MoS_2_/CT and CT the proportion of acrosome-reacted spermatozoa was similar among the groups, showing no significative differences (*p* > 0.05). Thus, it is possible to affirm that MoS_2_/CT and CT at the different concentrations tested have no statistically significant effect on acrosome integrity.

### 2.3. MoS_2_/CT and CT Supplementation Does Not Modify the Intracellular Calcium Concentration, Membrane Fluidity and Mitochondrial Activity

Flow cytometry analysis was performed to evaluate the effects of MoS_2_/CT and CT supplementations in three different events of sperm capacitation after 1.5 h of incubation in capacitating conditions: the increase in the intracellular calcium concentration; the sperm membrane disorder (thus revealing the potential membrane fluidity increase); and the mitochondrial activity. The results showed no differences in terms of intracellular calcium concentration, membrane disorder and mitochondrial activity, as graphically illustrated in Figure 3, Figure 4 and Figure 5, respectively. Appendix A shows the CTRL group after 0, 1.5 of capacitation.

Capacitation status has been associated with changes in the sperm machinery at membrane and cytosolic levels. For instance, the intracellular calcium ([Ca^2+^]^i^) homeostasis is a key element in sperm capacitation and acrosome reaction [31]. In resting conditions, the calcium clearance is tightly regulated by Ca^2+^ pumps (Ca^2+^-ATPases), Na^+^/Ca^2+^-exchangers, and Ca^2+^-channels in the sperm plasma membrane and in intracellular organelles, including the acrosome, the redundant-nuclear-envelope (RNE) and the mitochondria [32]. A key role is attributed to the Catsper channels, which are directly involved in sperm capacitation producing an influx of Ca^2+^ as a consequence of their stimulation [33,34]. During capacitation, the [Ca^2+^]^i^ rises and acts like a second messenger converting extracellular stimuli into a chemical response involving several molecular systems, such as, protein kinase C (PKC), protein kinase A (PKA), actin, and many others [35]. At the same time, one of most relevant events of AR is the very fast surge of [Ca^2+^]^i^, following the spermatozoa’s interaction with the oocyte [35]. Thus, Ca^2+^ is not only a homeostatic factor and a second messenger in spermatozoa, but it also controls and modulates the crucial physiological function in a sperm’s life, as also happens in excitable cells such as neurons, myocardiocytes, and muscular cells.

During sperm capacitation, the membrane physicochemical characteristics change, which is an important aspect to consider since the sperm cytosol is virtually absent, thus implying a direct contact of the sperm plasma membrane with underlying structures [36]. The sperm head plasma membrane (PM) presents a different composition between the inner and outer leaflets and this asymmetry is established and maintained by several translocating enzymes [36,37]. During capacitation, the phospholipid asymmetry is reduced and the phospholipids move inward and outward according to their concentration gradient [38]. The lipid remodeling allows the removal of cholesterol from extracellular protein, which determines the increase in the ability of the sperm plasma membrane (PM) and the outer acrosome membrane (OAM) to fuse (fusogenicity), a prerequisite for the acrosome reaction [39].

Mitochondrial activity is required for several sperm functions [40]. In addition to contributing to generate ATP, mitochondria act as a hub between the generation of reactive oxygen species (ROS) and the activation of apoptosis-related pathways [41]. The result of the capacitation process is the acquisition of fertilizing ability or activating pro-apoptotic pathways.

Thus, the findings obtained showed no differences in terms of mitochondrial activity, membrane disorder and intracellular calcium concentration, implying that MoS_2_/CT and CT do not exert any affect in these events of sperm capacitation in vitro.

### 2.4. MoS_2_/CT and CT Supplementation Does Not Influence Sperm pKa Activity and Tyrosine Phosphorylation Patterns

Flow cytometry was used to analyze the early events of capacitation that lead to PKA activation and the subsequent tyrosine phosphorylation cascade, which were then analyzed by Western blot (WB). Sperm protein soluble adenylyl cyclase (sAC) is activated during capacitation, raising the intracellular pH [42]. As a result, PKA and cAMP levels increase [43] and cAMP binds to the PKA regulatory subunits, allowing the dissociation of the tetramer and the activation of the catalytic subunit and thus initiating a cascade of intracytoplasmic signaling events [44]. Once released, the catalytic subunits continue their function phosphorylating a wide range of substrates on the Ser/Thr residues, activating a variety of protein kinases and/or inhibiting protein phosphatases to increase the phosphorylation of tyrosine residues either directly or indirectly [30,45] and contributing to reshaping the global protein phosphorylation pattern. This change occurs either in the flagellum or in the sperm head, and it appears to be mandatory for a spermatozoon to reach the ability to fertilize the oocyte [30,45]. While the results show the normal differences between non-capacitated and capacitated spermatozoa in terms of residues’ phosphorylation, our results showed no significant differences among the experimental groups. The phosphorylation patterns obtained were similar for MoS_2_/CT and CT (10, 1 and 0.1 ppm) with respect to the control group (Figure 6). All full-length membranes and immunoblotting images from three independent experiments were added to Appendix A.

### 2.5. CT Supplementation Improves the IVF Outcomes

Finally, IVF was used as a functional test to evaluate the sperms’ acquisition of the fertilizing ability after the exposure to MoS_2_/CT and CT at selected concentrations. Since the analysis of the PKA activity, phosphorylation patterns, acrosome reaction induction, mitochondrial activity, membrane fluidity and intracellular calcium concentration revealed no differences among the groups treated with different concentrations of MoS_2_/CT and CT (10, 1, 0.1 ppm), only the lowest concentrations were selected to perform the IVF assay, in order to minimize any potential risk, as well as the biological material needed.

IVF assays are a valuable tool to assess the sperm fertilizing ability in boar [46,47] due to the fact that the fertilization rates, the number of polyspermic oocytes and number of spermatozoa/polyspermic oocytes are related to the capacitation status and fertility of the semen [48,49]. As observed in Figure 7, the addition of CT alone at a specific concentration (0.1 ppm) during capacitation increases the spermatozoa fertilizing ability, evidenced by the increased number of fertilized oocytes and of polyspermic oocytes with respect to the control group.

Our findings support the evidence from previous studies showing that the addition of CT to the extender had a positive effect on sperm motility of caprine sperm [11].

This fact allows hypothesizing that the addition of CT might have provided an antioxidant activity, modulating the presence of reactive oxygen species (ROS) in our system. However, further experiments are needed to decipher the lipoperoxidation level and the ROS production, among others. The oxidative stress, observed by the excess of ROS, plays a key role in the life of mammalian sperm. Considering that spermatozoa are very sensitive to oxidative processes and that they are unable to transcribe and synthesize new proteins, with a cytosol virtually absent, preventing or fighting the oxidative stress remains hard to achieve [50]. This situation is aggravated due to the lipid composition of the plasma membrane, where the concentration of polyunsaturated fatty acids (PUFAs) is higher than in other cell types. PUFAs are the main target of ROS, and their oxidation culminates in the generation of cytotoxic aldehydes. Furthermore, the peroxidation of membrane lipids leads to a loss of motility and flexibility, causing the loss of all membrane-dependent functions [50,51,52]. The major consequence of oxidative stress is the damage of the sperm’s DNA [50,51].

Even if sperm cells are very susceptible to oxidative stress, low levels of ROS contribute to the full maturation of spermatozoa [50], which is necessary for capacitation, hyperactivation, acrosome reaction, oocyte fusion and fertilization [53,54]. Furthermore, there is a leading role in the interaction between ROS and cholesterol, since the oxidation of a part of cholesterol by ROS leads to the formation of oxysterols that facilitate the removal of cholesterol from the sperm plasma membrane to enhance the sperm membrane fluidity [55,56].

This study focused on the effect of an innovative material and a naturally derived compound on the spermatozoa fertilizing ability. Due to the results obtained, it is possible to affirm that nanoflakes of MoS_2_/CT at different concentrations (10, 1, 0.1 ppm) do not induce any negative effect on the sperm parameters related to capacitation evaluated here. Moreover, CT alone at a specific concentration (0.1 ppm) acts as a helper of sperm capacitation by improving the IVF results, which could probably be explained by the balance of the ROS levels. These results open encouraging new perspectives for the improvement of Assisted Reproduction Technologies (ART), which has experienced an increase in use during the last decades, both in humans [57] and in animal farming [58]. At the same time, an improvement in these techniques could fight against the ever increasing prevalence of numerous pathologies associated with ART-generated embryos, since ARTs may epigenetically modify gene expression, influencing the long-term development of the embryo by mechanisms that should still be investigated. For instance, many reports have shown the increasing prevalence of Angelman syndrome (AS), [59,60] and the phenomenon of large offspring syndrome in farm animals [61], with a phenotypic similarity to Beckwith–Wiedemann Syndrome (BWS) in humans [61,62,63].

## 3. Materials and Methods

### 3.1. Chemicals

Unless otherwise stated, all the chemicals were purchased from Sigma-Aldrich (St. Louis, MO, USA). Milli-Q water was purchased from Millipore (Bedford, MA, USA).

### 3.2. MoS_2_ Sonochemical Exfoliation in Water Assisted by Catechin

MoS_2_ exfoliation assisted by catechin (CT) was conducted according to Rojas and colleagues (2022) [12] with some modifications. A total of 0.5 g of bulk MoS_2_ powder (<2 µm; 99% purity) was placed in 50 mL of a 2.5 mg mL^−1^ (+)-catechin water solution (Milli-Q water) and roughly dispersed using a low-power ultrasonic bath. The dispersion was placed in a steel beaker and subjected to a sonochemical exfoliation treatment using a Branson SFX550 (550 W, 20 kHz) sonifier. The sonication probe employed was a 13 mm Ø Branson disruptor horn. Sonochemical exfoliation was conducted at 50% amplitude using a 5 h pulse program (2 s ON and 1 s OFF), maintaining the temperature below 15 °C. The obtained MoS_2_/CT dispersion was thus subjected to purification and size selection via differential centrifugation. The MoS_2_ not properly exfoliated was precipitated with centrifugation at 250× *g* (1 h). In this case, the supernatant was recovered, and the formed pellet discarded. The MoS_2_/CT size selection and removal of the exfoliation medium containing the residual catechin in solution were conducted on the supernatant of the previous treatment, achieved via a 20,000× *g* centrifugation (15 min). Then, the supernatant was removed, and the sediment collected and resuspended in water. The exfoliation yield was estimated via gravimetry, and MoS_2_/CT water dispersion and was stored at 4 °C in the dark. MoS_2_/CT before use was further purified via a 10,000× *g* centrifugation, where the supernatant was discarded, and the sediment was recovered in sterile Dulbecco-PBS to obtain a 200 mg L^−1^ MoS_2_-CT working dispersion.

The pristine and exfoliated MoS_2_ was characterized using a high-resolution scanning electron microscopy ΣIGMA (Carl Zeiss Microscopy GmbH, Munich, Germany). Solutions and materials used for in vitro studies were sterilized before their use and handled under a sterile hood. Catechin solutions were freshly prepared before use.

### 3.3. Experimental Groups

For the present work, a total of seven experimental groups were analyzed. Control group without CT nor MoS_2_/CT was included (CTRL). Complexes of MoS_2_ functionalized with catechin (MoS_2_/CT) were used at three different concentrations: MoS_2_/CT 10 ppm; MoS_2_/CT 1 ppm; and MoS_2_/CT 0.1 ppm. These concentrations were stablished based on the available literature [33] and the absence of a consensus regarding the use of specific concentrations. As control, catechin alone (CT) was utilized at the same concentrations (CT 10 ppm, CT 1 ppm, CT 0.1 ppm). Finally, Figure 8 schematically illustrates the experimental design, representing the experimental groups and the analyses performed.

### 3.4. Spermatozoa Preparation and Incubation

Spermatozoa were collected and washed following an already standardized protocol [64]. Briefly, sperm samples purchased from a specialized company (Società Agricola Geneetic S.r.l, Castellazzo, Italy) were incubated in a TCM199 medium supplemented with 13.9 mM glucose, 1.25 mM sodium pyruvate, 2.25 mM calcium lactate and 1 mM caffeine used to induce capacitation in vitro. MoS_2_/CT and CT dispersions at different concentrations were added to the capacitating medium to obtain the sample groups: CT 10 ppm, CT 1 ppm, CT 0.1 ppm, MoS_2_/CT 10 ppm, MoS_2_/CT 1 ppm, and MoS_2_/CT 0.1 ppm. Sperm cells were incubated at a final concentration of 1 × 10^7^ cells/mL for 0 or 1.5 h at 38.5 °C in 5% CO_2_ and a humidified atmosphere (Heraeus, Hera Cell). Sperm motility was visually estimated before the capacitation by light microscopy before each experiment and only samples with sperm motility > 90% were considered for further analyses.

### 3.5. Monitoring MoS_2_/CT Toxicity on Acrosome Integrity

Acrosome integrity was monitored by using a two stain technique with Hoechst 33,258 and FITC-PSA able to identify alive unreacted and reacted spermatozoa [65]. At least 100 cells were assessed by fluorescence microscopy in three independent experiments performed at different capacitation times (T0, T1.5), in the following groups: CTRL, MoS_2_/CT, and CT treated spermatozoa (10, 1, 0.1 ppm).

### 3.6. Flow Cytometry Analysis of Intracellular Calcium Concentration, Membrane Fluidity and Mitochondrial Activity

Flow cytometry analysis was performed to evaluate the differences between the different sperm groups in terms of: (a) intracellular calcium concentration; (b) sperm membrane lipid disorder; and (c) sperm mitochondrial activity. For each experiment and each condition (MoS_2_/CT 10 ppm; MoS_2_/CT 1 ppm; and MoS_2_/CT 0.1 ppm; CT 10 ppm; CT 1 ppm; CT 0.1 ppm; CTRL), three different biological and technical experiments were performed at 0 and 1.5 h of capacitation.

After capacitation, sperm cells were placed in a flow cytometry tube and incubated at RT while gently shaking with: (a) 1 µM Fluo 4-AM (15 min) to study the intracellular calcium concentration; (b) 1 µM DilC-12 (15 min) to measure the membrane lipid disorder; and (c) 1 µM Mitotracker Red (30 min) to check the activation of the mitochondria. To distinguish dead and live spermatozoa, two different stains were used when possible, depending on the fluorescence emission spectra of the different probes previously stated: 1 µM PI (5 min) was used in combination with Fluo 4-AM, while 1 µL LIVE/DEAD™ Fixable Near-IR Dead Cell Stain Kit (Catalog number: L10119, TermoFisher Scientific, Waltham, MA, USA) (10 min) was combined with Mitotracker Red. After the incubation time, 100,000 events/sample were acquired by flow cytometry (FACSCanto, BD Biosciences, Franklin Lakes, NJ, USA—three laser, eight color configuration). Each reagent was titrated (8-point titration) under assay conditions; dilutions were established based on achieving the highest signal (mean fluorescence intensity, MFI) for the positive population and the lowest signal for the negative population, representing the optimal signal-to-noise ratio, and stain indexes were calculated. Instrument performances, data reproducibility, and fluorescence calibrations were sustained and checked by the Cytometer Setup and Tracking Beads (BD Biosciences). To evaluate non-specific fluorescence, the Fluorescence Minus One (FMO) control was used. Compensation was assessed using CompBeads and FACSuite FC Beads (BD Biosciences) and single stained fluorescent samples. Data were analyzed first using FACSuite v 1.0.5 (BD Biosciences) software, and then FcsWizard Software was used to convert .fcs data to .csv format in order to perform an exhaustive analysis of the fluorescence emitted from every single spermatozoon for the various fluorescence probes [66]. To that, the columns “FCS”, “SSC”, “Fluo-4AM”, “PI”, “DilC12”, “Mito” and “NIR” with the data from the 100,000 events acquired, were selected and filtered following these criteria: forward scatter (FCS) between 35,000 and 135,000 arbitrary units (a.u); side scatter (SSC) between 20,000 and 145,000 a.u; Fluo 3-AM, M540 and Mitotracker Red >0 a.u; PI between 0 and 30,000 a.u; and near infrared between 0 and 20,000 a.u. Then, data were treated and subdivided in intervals of fluorescence intensity as follow: 41 intervals for intracellular calcium (from 0 to 20,000 a.u, 500 a.u range); 101 intervals for membrane disorder (from 0 to 50,000 AU, 500 a.u range); 61 intervals for mitochondrial activity (from 0 to 30,000 a.u, 500 a.u range). At least 98% of the data fitted within this range.

### 3.7. Evaluation of Sperm PKA Activity and Tyrosine Phosphorylation Patterns (pTyr) by Western Blot

To evaluate protein kinase A (PKA) activity and protein tyrosine phosphorylation pattern (pTyr), at 1.5 h of capacitation, sperm cells were diluted in a sample buffer 5× (5 mM DDT, 2% SDS, 1 M Tris, 10% Glycerol and 0.1% Bromophenol blue), heated (100 °C for 5 min) and centrifuged (15,000× *g* for 10 min at 4 °C). Proteins were migrated on an SDS-PAGE 4–15% gradient gel (Mini-PROTEAN^®^ TGX™ Precast Protein Gels, BioRad, Hercules, CA, USA) and blotted on a nitrocellulose membrane using the Trans-Blot^®^ TurboTM Transfer System (BioRad, Steenvoorde, France). The membranes were stained with Ponceau S solution and scanned, then membranes were blocked for 1 h in 5% (*w/v)* milk powder diluted in TBS-T and incubated with anti-phospho-pKa antibody (Phospho-pKa Substrate (RRXS*/T*), dilution 1:5000, Rabbit mAb, Cell Signaling, Leiden, The Netherlands) in 5% (*w/v*) BSA (*w/v*) in TBS-T (gently shaking, 4 °C, overnight). After washing, membranes were incubated with secondary antibody anti-rabbit HRP (1:10,000, Santa Cruz Technology, Dallas, TX, USA) for 1 h. Peroxidase was revealed using the SuperSignal™ West Pico PLUS Chemiluminescent Substrate (ThermoFisher, Waltham, MA, USA) and the images were digitally captured using an Azure C300 (Chemiluminescent Western Blot Imaging System, Azure Biosystems, Dublin, CA, USA). Tyrosine phosphorylation was assessed on the same membranes by stripping the previous antibodies with Restore^TM^ Western Blot Stripping Buffer (ThermoFisher). After washing, membranes were blocked in 10% (*w*/*v*) bovine gelatin (*w/v*) in TBS-T for 1 h and incubated with anti-phosphotyrosine antibody (Clone 4G10, dilution 1:10,000, Mouse mAb, Merck Millipore, Burlington, MA, USA) in PBS-T for 1.5 h. After washing, membranes were finally incubated with the secondary anti-Mouse HRP (1:10,000, Santa Cruz Technology) antibody for 1 h and revealed as described previously in this section. At least three biological replicates were performed for each antibody and experimental group.

### 3.8. In Vitro Fertilization Assay

To study the potential effects of MoS_2_/CT and CT on spermatozoa fertilizing ability, an in vitro fertilization (IVF) assay was carried out using an already validated protocol [65]. Ovaries from pre-pubertal gilts were collected at a local slaughterhouse and transported to the laboratory within 1 h at 25 °C. After washing, cumulus–oocyte complexes (COCs) were collected by aspirating the follicles that met the requirements (4–5 mm of diameter, translucent appearance, good vascularization and compactness of their granulosa layer and cumulus mass). Maturation was achieved in vitro by culturing the COCs in four-well dishes containing 500 μl of α-MEM medium supplemented with 10% FBS, 1% penicillin/streptomycin, 1% Ultraglutamine, 5 UI/mL hCG and 5 UI/mL PMSG for 44 h at 38.5 °C in a humidified atmosphere with 5% CO_2_ (Heraeus, Hera Cell, Hanau, Germany).

Once matured, oocytes were denuded in Dulbecco-PBS with hyaluronidase on a warmed stage at 38.5 °C under a stereomicroscope. Only oocytes presenting the first polar body (MII stage) under the stereomicroscope were used for the IVF assay. Matured oocytes and capacitated sperm cells (1 × 10^6^ cells/mL) from the selected groups (CTRL, CT 0.1 ppm and MoS_2_/CT 0.1 ppm) were co-incubated in a fertilization medium (capacitation medium supplemented with 10% FBS). After 3 h of co-incubation, oocytes were transferred to a fresh medium and maintained in culture for at least 12 h. The penetration rate was evaluated after staining with Hoechst 33,342 and assessed under the fluorescence microscope. The IVF outcomes are expressed as fertilization rate (% of penetrated oocytes), incidence of polyspermy (% of polyspermic oocytes) and number of penetrating spermatozoa/polyspermic oocyte according to already published and valuable works [47,65]. A total of four independent experiments were performed, reaching a total number of 104 oocytes (number of fertilized oocytes per group: CTRL, 10 fertilized of 33 total oocytes; CT 0.1 ppm 18 fertilized of 34 total oocytes; MoS_2_/CT 0.1 ppm 18 fertilized of 37 total oocytes).

### 3.9. Statistical Analysis

For statistical analysis, GraphPad Prism 6 Software (La Jolla, CA, USA) was used. Data were checked for normal distribution with a D’Agostino and Pearson normality test prior to performing the comparison with parametric or non-parametric tests, as required. In all cases, the differences among groups were considered statistically significant when *p* < 0.05. To normalize the western bot data, Ponceau red staining was used, following a validated protocol [67]. Briefly, the whole lanes were quantified by densitometry and bands were afterwards quantified using ImageQuantTL (GE Healthcare LifeSciences, Barrington, IL, USA). To assess the effect of different treatments on IVF, four independent technical and biological experiments were carried out. An a priori power analysis was performed to establish the number of oocytes with G*Power 3.1.9.7 software, obtaining a final power of our analysis ≥95%.

## 4. Conclusions

In conclusion, our study demonstrates that the incubation of spermatozoa in the presence of catechins (0.1 ppm) enhances their fertilizing ability and the incubation with nanoflakes of MoS_2_/CT at different concentrations do not induce any negative effect on the sperm parameters related to capacitation. However, further experiments are needed to decipher the exact mechanism by which catechins are able to increase the sperm fertilizing ability and to explore their antioxidant potential on spermatozoa.

The findings open interesting perspectives regarding the use of catechins and new materials obtained using natural or bio compounds, which could be used to implement the current strategies for sperm capacitation.

## Figures and Tables

**Figure 1 ijms-24-04788-f001:**
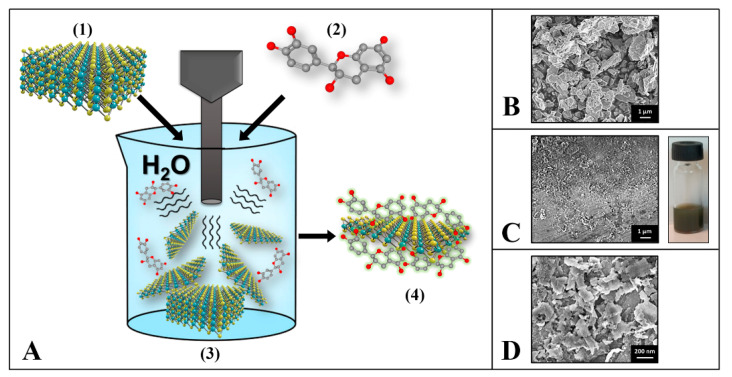
(**A**) Graphical sketch of the MoS_2_ sonochemical exfoliation assisted by catechin. (**1**) Bulk MoS_2_, (**2**) catechin structure, (**3**) sonochemical exfoliation process, (**4**) MoS_2_/CT nanoflakes. (**B**) SEM micrograph of the bulk-MoS_2_ (unexfoliated); (**C**) SEM micrograph of the exfoliated MoS_2_/CT (**left**); picture of the MoS_2_-CT colloidal water-dispersion (**right**). (**D**) SEM magnification of the MoS_2_/CT nanoflakes.

**Figure 2 ijms-24-04788-f002:**
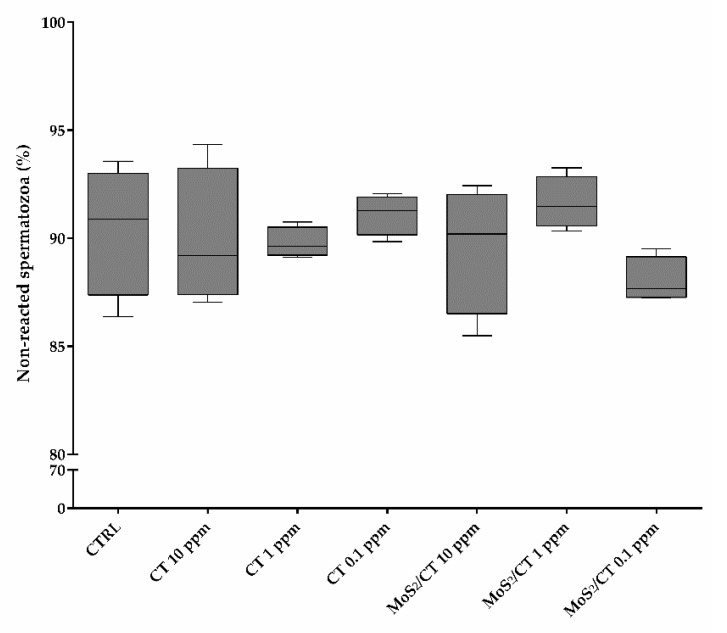
Acrosome integrity. The graph shows the percentage of non-reacted spermatozoa after 1.5 h of capacitation for the different experimental groups: CTRL, CT 10 ppm, CT 1 ppm, CT 0.1, MoS_2_/CT 10 ppm, MoS_2_/CT 1 ppm and MoS_2_/CT 0.1 ppm. A normal acrosome damage rate was obtained, similar to the control (CTRL) group (*p* > 0.05). Three independent technical and biological (from different boars) replicates were performed.

**Figure 3 ijms-24-04788-f003:**
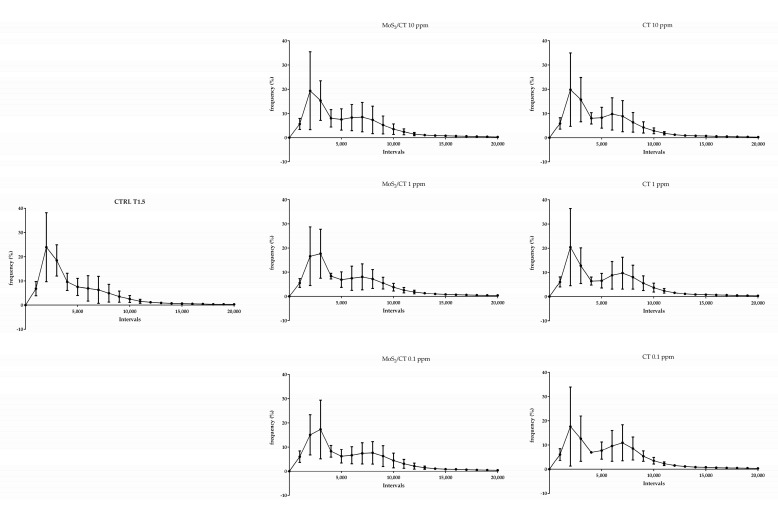
Flow cytometry analysis of intracellular calcium concentration. The graphs show the frequency of spermatozoa emitting a specific fluorescence intensity, which was subdivided into intervals ranging from 0 to 20,000 a.u. Capacitation was performed up to 1.5 h. Fluo 4-AM was used in combination with PI. Three independent technical and biological experiments were performed (*n* = 3).

**Figure 4 ijms-24-04788-f004:**
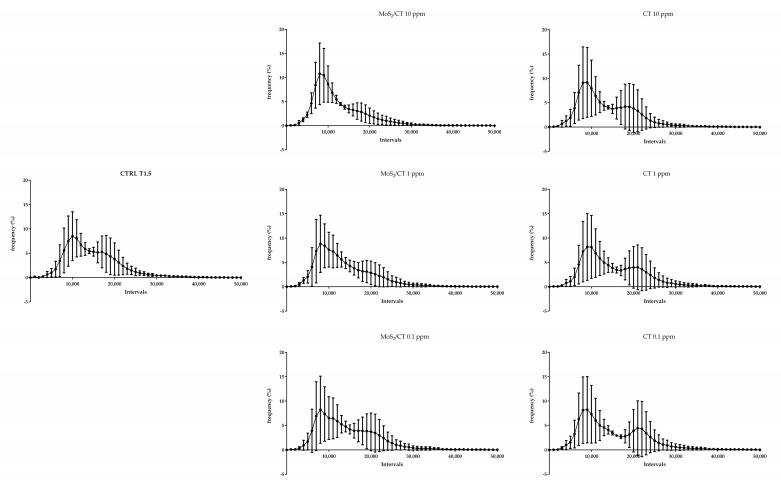
Flow cytometry analysis of membrane disorder and fluidity. The graphs show the frequency of spermatozoa emitting a specific fluorescence intensity, which was subdivided into intervals ranging from 0 to 50,000 a.u. Capacitation was performed up to 1.5 h. DilC-12 was used as a probe. Three independent technical and biological experiments were performed (*n* = 3).

**Figure 5 ijms-24-04788-f005:**
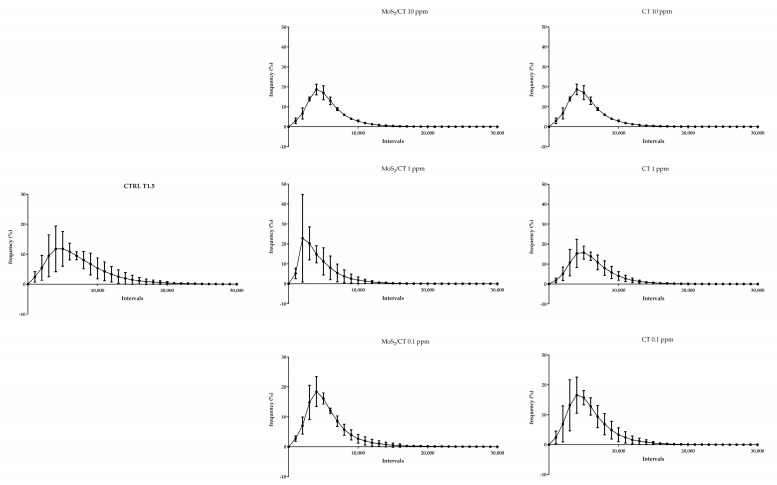
Flow cytometry analysis of mitochondrial activity. The graphs show the frequency of spermatozoa emitting a specific fluorescence intensity, which was subdivided into intervals ranging from 0 to 30,000 a.u. Capacitation was performed up to 1.5 h. Mitotracker Red was used in combination with a near-IR probe. Three independent technical and biological experiments were performed (*n* = 3).

**Figure 6 ijms-24-04788-f006:**
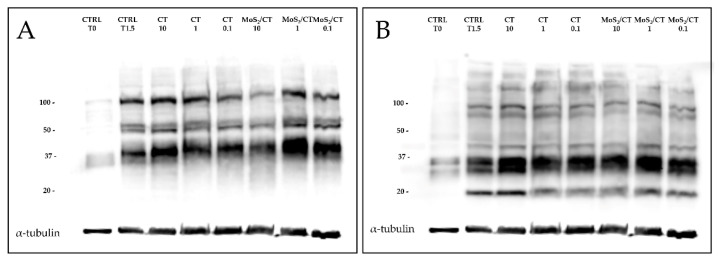
Western blot analysis of PKA activity and tyrosine phosphorylation patterns. The image illustrates (**A**) the PKA activity and (**B**) the tyrosine phosphorylation patterns after 1.5 h of incubation under capacitating conditions. Antibodies were incubated on the same blot after membrane stripping and re-blotting. Blots were cut prior to hybridization. α-tubulin was used as a load control. Each of the eight lanes contains 1 × 10^7^ spermatozoa from different animals. At least three independent experiments with different animals were performed.

**Figure 7 ijms-24-04788-f007:**
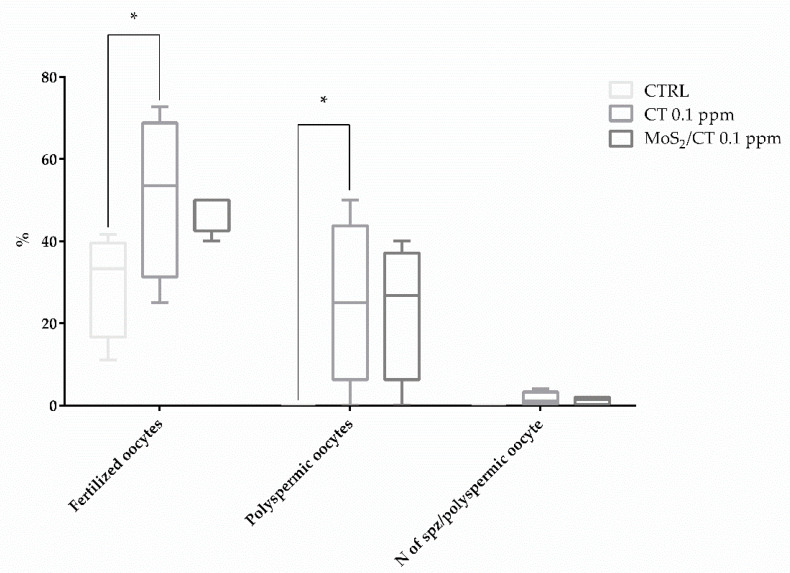
IVF outcomes. Three different groups were subjected to IVF assay: CTRL, CT 0.1 and MoS_2_/CT 0.1 ppm. Data are expressed as percentages, showing the number of fertilized oocytes, the number of polyspermic oocytes and the number of spermatozoa per polyspermic oocyte, comparing the groups of spermatozoa capacitated in the presence of MoS_2_/CT and CT (0.1 ppm) to the control group. Data were analyzed using Dunnett’s test. * *p* < 0.05 versus control. Four independent experiments were performed.

**Figure 8 ijms-24-04788-f008:**
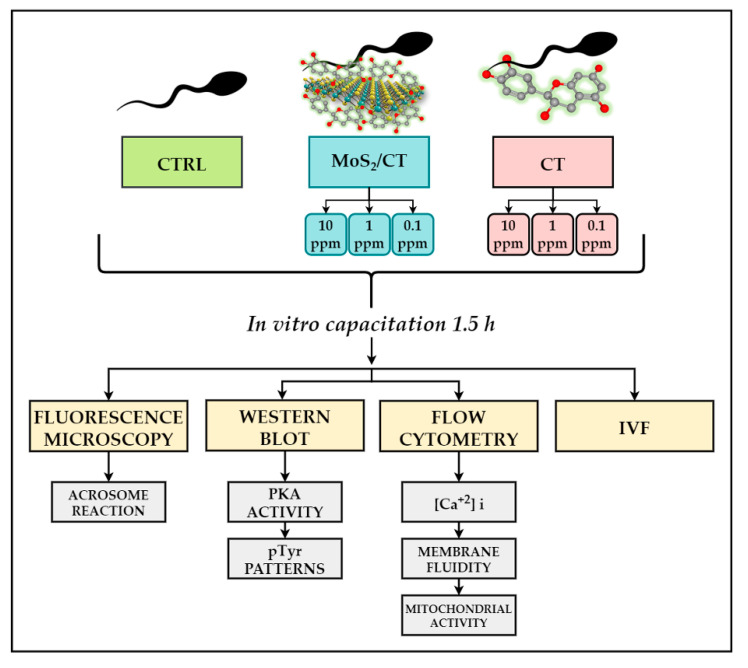
Experimental design. Spermatozoa were exposed to MoS_2_ functionalized with catechin at different concentrations (10, 1, 0.1 ppm), catechin alone at the same concentrations (10, 1, 0.1 ppm) for 1.5 h in capacitation medium, a control group (CTRL) was maintained. Different sperm capacitation events were analyzed: acrosome damage, membrane fluidity, mitochondrial activity, intracellular calcium concentration, biochemical phosphorylation patterns and IVF assays.

## Data Availability

The original contributions presented in the study are included in the article/Appendix A, further inquiries can be directed to the corresponding author.

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
