# Peer review of "Catechin versus MoS2 Nanoflakes Functionalized with Catechin: Improving the Sperm Fertilizing Ability—An In Vitro Study in a Swine Model"

_ijms, 2023, doi:10.3390/ijms24054788_

Round 1

Reviewer 1 Report

Catechin versus MoS2 nanoflakes functionalized with catechin: 2 improving the sperm fertilizing ability – an in vitro study on 3 swine model.

Thanks to Costanza Cimini et al. for this study. Before moving forward there are some questions and errors to be corrected.

Q1. Authors topic is related to sperm capacitation & in vitro fertilization. Therefore, brief definition of these topics sperm capacitation & IVF should be added in introduction section. It is not necessary that research papers are read by researchers or physicians in related topics. It can be read by others also.

Q2. Results and Discussion should be separated. Separate section allows reader to more easily understand the findings of the research and interpret the findings.  

Q3. Line 233, write full form of WB if once used in manuscript.

Q4. In figure 6, which band is PKA and tyrosine phosphorylation is not clear, there are 3 bands which one should we consider? crop the image and use proper name of antibodies with its corresponding molecular weight. The current image used can be used as supplemental with its molecular weight mentioned along with marker. Also make a graph analysis of Western blot and compare it with your control alpha-tubulin.

Q5. Line 332, “25O g” what is “O”? Is it numerical “zero” or alphabet “O”?

Q6. Catsper is sperm specific, Ca++ permeable channel essential for hyperactivity of sperm flagellum, chemotaxis towards eggs, capacitation & acrosome reaction. All these physiological events require calcium entry in sperm cells. Therefore, Authors should do western blot of Catsper to support your findings. Also authors should add about catsper in introduction, result and discussion.

Author Response

The Authors thank the Editor and Referees for their careful and rapid review of our manuscript. All the modifications suggested by the Referees have been appreciated and followed, and we consider that the manuscript has considerably improved now. Please find enclosed a clean version of the revised manuscript (IJMS_cleanversion) and a file with all the highlighted modifications (IJMS _highlighted). We really hope that you could reconsider our new version of the manuscript as suitable for publication in IJMS.

We remain at your disposal in case you need any other clarification or additional information.

Looking forward to hearing from you, we send you our best regards.

Reviewers’ comments:

Reviewer #1. Authors topic is related to sperm capacitation & in vitro fertilization. Therefore, brief definition of these topics sperm capacitation & IVF should be added in introduction section. It is not necessary that research papers are read by researchers or physicians in related topics. It can be read by others also.

Reply: We thank Reviewer 1 for her/his careful reading and for the comments and suggestions. Accordingly, we have added an introduction on sperm capacitation and IVF (Please see L37-40).

Q2. Results and Discussion should be separated. Separate section allows reader to more easily understand the findings of the research and interpret the findings.  

Reply: Thank you for your proposal. Although we consider your comment very valuable, and we have agreed with you in other manuscripts, we believe that, in this specific case, results and discussions should be merged, since it promotes the work done and makes understanding of the results more immediate, following a fluid timeline. We apologize for the inconveniences.

Q3. Line 233, write full form of WB if once used in manuscript.

Reply: We apologize for the mistake; we have corrected the sentence (Please see L236).

Q4. In figure 6, which band is PKA and tyrosine phosphorylation is not clear, there are 3 bands which one should we consider? crop the image and use proper name of antibodies with its corresponding molecular weight. The current image used can be used as supplemental with its molecular weight mentioned along with marker. Also make a graph analysis of Western blot and compare it with your control alpha-tubulin.

Reply: Thank you for your comments. In this study, we did not evaluate single bands, but the activity of an important protein such as PKA, as well as a pattern of tyrosine phosphorylation. For this reason, we did not crop the membranes. All the antibodies used are specified within the Materials and Methods section. Moreover, the scope of this kind of analysis is not to “quantify” (considering the limits of WB technique, since it represents a semi-quantitative analysis) the protein expression, but to confirm the absence of differences in the activity of a protein and in the phosphorylation patterns of a protein cocktail. Here below are some interesting manuscripts of very experts in the field:

Soriano-Úbeda, C., Romero-Aguirregomezcorta, J., Matás, C. et al. Manipulation of bicarbonate concentration in sperm capacitation media improves in vitro fertilisation output in porcine species. J Animal Sci Biotechnol 10, 19 (2019). https://doi.org/10.1186/s40104-019-0324-y

Visconti PE, Moore GD, Bailey JL, Leclerc P, Connors SA, Pan D, et al. Capacitation of mouse spermatozoa. II. Protein tyrosine phosphorylation and capacitation are regulated by a cAMP-dependent pathway. Development. 1995;121(4):1139–50 Available from: http://dev.biologists.org/content/121/4/1139.full.pdf.

Q5. Line 332, “25O g” what is “O”? Is it numerical “zero” or alphabet “O”?

Reply: We apologize for the mistake; we have corrected the sentence (Please see L336). It corresponds to 250 x g.

Q6. Catsper is sperm specific, Ca++ permeable channel essential for hyperactivity of sperm flagellum, chemotaxis towards eggs, capacitation & acrosome reaction. All these physiological events require calcium entry in sperm cells. Therefore, Authors should do western blot of Catsper to support your findings. Also, authors should add about catsper in introduction, result and discussion.

Reply: The physiology of calcium control is very complex in spermatozoa either in resting conditions as well as during capacitation and acrosome reaction. Indeed, Ca2+ intracellular concentration is one of the key modulators of sperm function and the most important signaling system, involved in virtually all the biochemical processes involved in capacitation.

It involves thousands of different kinds of channels and pumps, see for references:

Bernabò, N., Palestini, P., Chiarini, M., Maccarrone, M., Mattioli, M., Barboni, B.Endocannabinoid-binding CB1 and TRPV1 receptors as modulators of sperm capacitation (2012) Communicative and Integrative Biology, 5 (1), pp. 68-70.

Botto, L., Bernabò, N., Palestini, P., Barboni, B. Bicarbonate induces membrane reorganization and CBR1 and TRPV1 endocannabinoid receptor migration in lipid microdomains in capacitating boar spermatozoa (2010) Journal of Membrane Biology, 238 (1-3), pp. 33-41.

Bernabò, N., Pistilli, M.G., Mattioli, M., Barboni, B. Role of TRPV1 channels in boar spermatozoa acquisition of fertilizing ability (2010) Molecular and Cellular Endocrinology, 323 (2), pp. 224-231.

Bernabò, N., Mattioli, M., Barboni, B. The spermatozoa caught in the net: The biological networks to study the male gametes post-ejaculatory life (2010) BMC Systems Biology, 4, art. no. 87, .

For this reason, we concentrated our attention on the final event (the variation in Calcium concentration) instead that on the involved molecules, that are a myriad. See for references:

Lacalle E, Consuegra C, Martínez CA, Hidalgo M, Dorado J, Martínez-Pastor F,Álvarez-Rodríguez M, Rodríguez-Martínez H. Bicarbonate-Triggered In Vitro Capacitation of Boar Spermatozoa Conveys an Increased Relative Abundance of the Canonical Transient Receptor Potential Cation (TRPC) Channels 3, 4, 6 and 7 and of CatSper-γ Subunit mRNA Transcripts. Animals (Basel). 2022 Apr 13;12(8):1012. doi: 10.3390/ani12081012. PMID: 35454259; PMCID: PMC9031844.

Yeste M, Fernández-Novell JM, Ramió-Lluch L, Estrada E, Rocha LG, Cebrián- Pérez JA, Muiño-Blanco T, Concha II, Ramírez A, Rodríguez-Gil JE. Intracellular calcium movements of boar spermatozoa during 'in vitro' capacitation and subsequent acrosome exocytosis follow a multiple-storage place, extracellular calcium-dependent model. Andrology. 2015 Jul;3(4):729-47. doi:10.1111/andr.12054. Epub 2015 Jun 20. PMID: 26097097.

Macías-García B, García-Marín LJ, Bragado MJ, González-Fernández L. The calcium-sensing receptor regulates protein tyrosine phosphorylation through PDK1 in boar spermatozoa. Mol Reprod Dev. 2019 Jul;86(7):751-761. doi:

10.1002/mrd.23160. Epub 2019 May 9. PMID: 31074040.

 Vicente-Carrillo A, Álvarez-Rodríguez M, Rodríguez-Martínez H. The CatSperchannel modulates boar sperm motility during capacitation. Reprod Biol. 2017 Mar;17(1):69-78. doi: 10.1016/j.repbio.2017.01.001. Epub 2017 Jan 8. PMID: 28077244.

Harayama H. Roles of intracellular cyclic AMP signal transduction in the capacitation and subsequent hyperactivation of mouse and boar spermatozoa. J Reprod Dev. 2013 Oct;59(5):421-30. doi: 10.1262/jrd.2013-056. PMID: 24162806;PMCID: PMC3934125.

Vadnais ML, Galantino-Homer HL, Althouse GC. Current concepts of molecular events during bovine and porcine spermatozoa capacitation. Arch Androl. 2007 May-Jun;53(3):109-23. doi: 10.1080/01485010701329386. PMID: 17612869.

Dubé C, Tardif S, LeClerc P, Bailey JL. The importance of calcium in the appearance of p32, a boar sperm tyrosine phosphoprotein, during in vitro capacitation. J Androl. 2003 Sep-Oct;24(5):727-33. doi: 10.1002/j.1939-4640.2003.tb02734.x. PMID: 12954665.

Shadan S, James PS, Howes EA, Jones R. Cholesterol efflux alters lipid raft stability and distribution during capacitation of boar spermatozoa. Biol Reprod. 2004 Jul;71(1):253-65. doi: 10.1095/biolreprod.103.026435. Epub 2004 Mar 17. PMID: 15028630.

Martínez-Abad S, Castillo-Martín M, Gadani B, Rodríguez-Gil JE, Bonet S, Yeste M. Voltage-dependent anion channel 2 is involved in in vitro capacitation of boar sperm. Reprod Domest Anim. 2017 Oct;52 Suppl 4:65-68. doi: 10.1111/rda.13060. PMID: 29052329.

Mata-Martínez E, Sánchez-Cárdenas C, Chávez JC, Guerrero A, Treviño CL, Corkidi G, Montoya F, Hernandez-Herrera P, Buffone MG, Balestrini PA, Darszon A. Role of calcium oscillations in sperm physiology. Biosystems. 2021 Nov;209:104524. doi: 10.1016/j.biosystems.2021.104524. Epub 2021 Aug 26. Erratum in: Biosystems. 2021 Dec;210:104546. PMID: 34453988.

Li X, Wang L, Li Y, Zhao N, Zhen L, Fu J, Yang Q. Calcium regulates motility and protein phosphorylation by changing cAMP and ATP concentrations in boar sperm in vitro. Anim Reprod Sci. 2016 Sep;172:39-51. doi: 10.1016/j.anireprosci.2016.07.001. Epub 2016 Jul 2. PMID: 27423488.

Reviewer 2 Report

The research evaluated the effect of catechin and molybdenum disulphide on sperm capacitation and sperm fertilizing ability in an IVF assay.

The results are very interesting and confirm the hypothesis that the compound do not induce any negative effect on the parameters evaluated related to sperm capacitation. Furthermore, a positive effect of catechin on sperm fertilizing ability was found.

The introduction is clear and complete, clearly identifying the reasons behind the research.

  The methodology is carefully explained. The results are accurately described and discussed. As my only suggestion, I would ask the authors to comment on the increased polyspermy found in "in vitro" tests with addition of catechin and/or molybdenum disulphide. Polyspermy could prove to be a serious problem for future practical application.

Author Response

The Authors thank the Editor and Referees for their careful and rapid review of our manuscript. All the modifications suggested by the Referees have been appreciated and followed, and we consider that the manuscript has considerably improved now. Please find enclosed a clean version of the revised manuscript (IJMS_cleanversion) and a file with all the highlighted modifications (IJMS _highlighted). We really hope that you could reconsider our new version of the manuscript as suitable for publication in IJMS.

We remain at your disposal in case you need any other clarification or additional information.

Looking forward to hearing from you, we send you our best regards.

Reviewers’ comments:

Reviewer 2#. The research evaluated the effect of catechin and molybdenum disulphide on sperm capacitation and sperm fertilizing ability in an IVF assay.

The results are very interesting and confirm the hypothesis that the compound do not induce any negative effect on the parameters evaluated related to sperm capacitation. Furthermore, a positive effect of catechin on sperm fertilizing ability was found.

The introduction is clear and complete, clearly identifying the reasons behind the research.

  The methodology is carefully explained. The results are accurately described and discussed. As my only suggestion, I would ask the authors to comment on the increased polyspermy found in "in vitro" tests with addition of catechin and/or molybdenum disulphide. Polyspermy could prove to be a serious problem for future practical application.

Reply: We thank Reviewer 2 for her/his diligent reading, opinions and ideas. Thank you for your very appreciated comments. Here, we used the IVF as a functional test for spermatozoa, without the aim of obtaining embryos. It means that our work was set up to evaluate the capacitation status of sperm cells in a specific context: after exposure during capacitation to MoS2 functionalized with catechin and catechins alone. As it has been reported in literature, in swine, in vitro maturated oocytes are partially unable to avoid the polyspermic fertilization, probably due to the altered zona pellucida function and/or the inefficient extrusion of cortical bodies immediately after the spermatozoa entry in female gametes. Please, see for references:

Funahashi H. Polyspermic penetration in porcine IVM-IVF systems. Reprod Fertil Dev. 2003;15(3):167-77.

Ito J, Kashiwazaki N. Molecular mechanism of fertilization in the pig. Anim Sci J. 2012 Oct;83(10):669-82. 

Hayashi T, Sato H, Iwata H, Kuwayama T, Monji Y. Inhibitory effects of calcium ionophore pretreatment of porcine oocytes on polyspermic fertilization. Zygote. 2006 Feb;14(1):17-22.

This particular feature, in one hand, poses specific problems in management of in vitro IVM/IVF protocols in swine, while on the other hand, it allows to use the quantification of polyspermy as a valuable tool to measure the fertilizing ability of spermatozoa. In particular, it has been proposed that two parameters (% of polyspermyc oocytes, number of spermatozoa/polyspermyc oocyte) could be related with the capacitation status of male gametes. Please, see for references:

Abeydeera LR and Day BN. In vitro penetration of pig oocytes in a modified Tris-buffered medium: effect of BSA, caffeine and calcium. Theriogenology 1997;48:537–44.

Abeydeera LR and Day BN. Fertilization and subsequent development in vitro of pig oocytes inseminated in a modified Trisbuffered medium with frozen-thawed ejaculated spermatozoa. Biol Reprod 1997;57:729–34.

Suzuki, H.; Saito, Y.; Kagawa, N.; Yang, X. In vitro fertilization and polyspermy in the pig: Factors affecting fertilization rates and cytoskeletal reorganization of the oocyte. Microsc. Res. Tech. 2003, 61, 327–334, doi:10.1002/jemt.10345.

Bernabò N et al. Effects of 50 Hz extremely low frequency magnetic field on the morphology and function of boar spermatozoa capacitated in vitro. Theriogenology. 2007 Mar 1;67(4):801-15.

Bernabò N et al. Graphene oxide affects in vitro fertilization outcome by interacting with sperm membrane in an animal model. Carbon Volume 129, April 2018, Pages 428-437.

Cimini C et al. Pre-Treatment of Swine Oviductal Epithelial Cells with Progesterone Increases the Sperm Fertilizing Ability in an IVF Model. Animals (Basel). 2022 May 6;12(9):1191

Reviewer 3 Report

In this manuscript, the authors analyze the effect of a non-physiological materials, MoS2 and a flavonoid with antioxidant capacity, catechins (CT), on sperm fertility ability in pig. They analyzed 3 different concentrations of nanoflakes of MoS2 functionalized with CT and CT during sperm in vitro capacitation, and they found that they do not induce any negative effect on sperm capacitation (acrosome damage;  membrane disorder; biochemical patterns -PKA activity and tyrosine phosphorylation patterns-; intracellular calcium concentration; and mitochondrial activity), and that CT at the lower concentration, increased the sperm fertilizing ability in the IVF, producing a higher number of fertilized oocyte than the control.

The main problem of the work is that the authors propose analyze the effect of the 2 components (MoS2 and CT) in capacitation at 1.5 hours, and for a species like porcine, where fresh sperm lasts up to a week in vitro, it is a very short time to analyze spontaneously capacitation. In 1.5 hours, the medium used with caffeine does not produce any effect on the control, and also indicates that the medium does not contain FBS, something used for capacitation in the IVF experiment.

 In order to talk about capacitation, it should be induced experimentally in each treatment and see how it fluctuates over a longer period of time. In this manuscript, they have analyzed the effect on toxicity at 1.5 hours of these 2 components, not in capacitation. This must be clear in the manuscript.

I have some minor points that should be answered:

-CT because its antioxidant properties could have a benefit for sperm, but why analyze MoS2?, The authors should improve the argument for which they have analyzed this.

-line 157: Depend of the animal species that “spermatozoa that have lost their acrosome are unable to fertilize an egg”, in mice it has been demonstrated that sperm without acrosome can fertilize. Qualify the phrase.

-Fig 2: In material and methods the authors indicate that they analyse the status of the acrosome of living and dead, indicate what figure 2 corresponds to. Indicate if the biological replicas are from three different individuals or from three ejaculates (same for the rest of figures).

- Presenting 35% of fertilized oocytes is a very low figure, current IVF systems in pig hardly reach 50% to 60% efficiency, similar to that obtained with CT in this manuscript.

-Indicate the number of fertilized oocytes per group in the four independent experiments that indicate that they have performed.

-line 278, the authors say that their “findings support the evidences from previous studies showing that the addition 278 of CT to the extender had a positive effect on sperm motility of caprine sperm”.  In previous manuscripts the authors have reported 90% of fertilized oocytes (reference 63 used in M&M for indicated the protocol used in this work for IVF).

-Why has motility not been analysed in this work? Why has viability not been analysed if CT is related to oxidative stress? To analyze the toxicity of these components it is necessary to analyze motility and viability.

Author Response

In this manuscript, the authors analyze the effect of a non-physiological materials, MoS2 and a flavonoid with antioxidant capacity, catechins (CT), on sperm fertility ability in pig. They analyzed 3 different concentrations of nanoflakes of MoS2 functionalized with CT and CT during sperm in vitro capacitation, and they found that they do not induce any negative effect on sperm capacitation (acrosome damage;  membrane disorder; biochemical patterns -PKA activity and tyrosine phosphorylation patterns-; intracellular calcium concentration; and mitochondrial activity), and that CT at the lower concentration, increased the sperm fertilizing ability in the IVF, producing a higher number of fertilized oocyte than the control.

The main problem of the work is that the authors propose analyze the effect of the 2 components (MoS2 and CT) in capacitation at 1.5 hours, and for a species like porcine, where fresh sperm lasts up to a week in vitro, it is a very short time to analyze spontaneously capacitation. In 1.5 hours, the medium used with caffeine does not produce any effect on the control, and also indicates that the medium does not contain FBS, something used for capacitation in the IVF experiment.

 In order to talk about capacitation, it should be induced experimentally in each treatment and see how it fluctuates over a longer period of time. In this manuscript, they have analyzed the effect on toxicity at 1.5 hours of these 2 components, not in capacitation. This must be clear in the manuscript.

 Reply: swine in vitro capacitation has been extensively studied in last 20 years by different groups and the molecular events that characterize this process are relatively well understood. See references:

Harrison RA, Gadella BM. Bicarbonate-induced membrane processing in sperm capacitation. Theriogenology. 2005 Jan 15;63(2):342-51. doi: 10.1016/j.theriogenology.2004.09.016. PMID: 15626403.

Gadella BM, Harrison RA. Capacitation induces cyclic adenosine3',5'-monophosphate-dependent, but apoptosis-unrelated, exposure of aminophospholipids at the apical head plasma membrane of boar sperm cells. Biol Reprod. 2002 Jul;67(1):340-50. doi: 10.1095/biolreprod67.1.340. PMID: 12080038.

Gadella BM, Harrison RA. The capacitating agent bicarbonate induces protein kinase A-dependent changes in phospholipid transbilayer behavior in the sperm plasma membrane. Development. 2000 Jun;127(11):2407-20. doi: 10.1242/dev.127.11.2407. PMID: 10804182.

Brewis IA, Moore HD, Fraser LR, Holt WV, Baldi E, Luconi M, Gadella BM, Ford WC, Harrison RA. Molecular mechanisms during sperm capacitation. Hum Fertil (Camb). 2005 Dec;8(4):253-61. doi: 10.1080/14647270500420178. PMID: 16393825.

Leahy T, Gadella BM. Capacitation and capacitation-like sperm surface changes induced by handling boar semen. Reprod Domest Anim. 2011 Sep;46 Suppl 2:7-13. doi: 10.1111/j.1439-0531.2011.01799.x. PMID: 21884270.

Flesch FM, Colenbrander B, van Golde LM, Gadella BM. Capacitation induces tyrosine phosphorylation of proteins in the boar sperm plasma membrane. Biochem Biophys Res Commun. 1999 Sep 7;262(3):787-92. doi: 10.1006/bbrc.1999.1300. PMID:10471403.

Tsai PS, Gadella BM. Molecular kinetics of proteins at the surface of porcine sperm before and during fertilization. Soc Reprod Fertil Suppl. 2009;66:23-36.PMID: 19848264.

Purdy PH, Graham JK, Azevedo HC. Evaluation of boar and bull sperm capacitation and the acrosome reaction using flow cytometry. Anim Reprod Sci.2022 Nov;246:106846. doi: 10.1016/j.anireprosci.2021.106846. Epub 2021 Sep 11.PMID: 34563407.

Sutovsky P, Kerns K, Zigo M, Zuidema D. Boar semen improvement through sperm capacitation management, with emphasis on zinc ion homeostasis. Theriogenology.2019 Oct 1;137:50-55. doi: 10.1016/j.theriogenology.2019.05.037. Epub 2019 May31. PMID: 31235187.

Ded L, Dostalova P, Zatecka E, Dorosh A, Komrskova K, Peknicova J.

Fluorescent analysis of boar sperm capacitation process in vitro. Reprod BiolEndocrinol. 2019 Dec 19;17(1):109. doi: 10.1186/s12958-019-0554-z. PMID:31856844; PMCID: PMC6923987.

Ni F, Chenling G, Fang H, Xun L, Xiaoye W, Yinsheng T, Mingguang H, Chuanhuo H. Analysis of differential proteins between non-capacitated and capacitated boar sperm and verification of the effect of phosphofructokinase oncapacitation. Theriogenology. 2022 Dec 29;199:19-29. doi:10.1016/j.theriogenology.2022.12.038. Epub ahead of print. PMID: 36682265.

Wang Y, Gu Y, Gao H, Gao Y, Shao J, Pang W, Dong W. Exploring boar sperm sialylation during capacitation using boronic acid-functionalized beads.Reproduction. 2018 Jan;155(1):25-36. doi: 10.1530/REP-17-0369. PMID: 29269442.

Zigo M, Kerns K, Sutovsky M, Sutovsky P. Modifications of the 26S proteasome during boar sperm capacitation. Cell Tissue Res. 2018 Jun;372(3):591-601. doi: 10.1007/s00441-017-2786-6. Epub 2018 Jan 29. PMID: 29376192; PMCID: PMC5949253.

In this context, it is possible to distinguish early events that take form seconds to a few minutes to be detectable (e.g.,  the promotion of membrane disorder exerted by bicarbonate) to long lasting ones (e.g., Protein phosphorylation) that require several minutes to 1-2 hours to be completed. The fine modulation of these processes, of course, depends on the specific system and by the artificial environment used in the experiment.  In our model we obtain a plateau of about 25-35% of capacitated spermatozoa after 1 1.5 hours, that remains stable for about 2 hours. After 4 h of incubation under capacitating conditions the fertilizing ability of the sample start to decrease, while the percentage of apoptotic or ding spermatozoa will increase. See for references:

Bernabò N, Tettamanti E, Pistilli MG, Nardinocchi D, Berardinelli P, Mattioli M, Barboni B. Effects of 50 Hz extremely low frequency magnetic field on the morphology and function of boar spermatozoa capacitated in vitro.Theriogenology. 2007 Mar 1;67(4):801-15. doi:10.1016/j.theriogenology.2006.10.014. Epub 2006 Dec 29. PMID: 17196643.

Botto L, Bernabò N, Palestini P, Barboni B. Bicarbonate induces membrane reorganization and CBR1 and TRPV1 endocannabinoid receptor migration in lipid microdomains in capacitating boar spermatozoa. J Membr Biol. 2010 Dec;238(1-3):33-41. doi: 10.1007/s00232-010-9316-8. Epub 2010 Nov 23. PMID:

21104238.

Barboni B, Bernabò N, Palestini P, Botto L, Pistilli MG, Charini M, Tettamanti E, Battista N, Maccarrone M, Mattioli M. Type-1 cannabinoid receptors reduce membrane fluidity of capacitated boar sperm by impairing their activation by bicarbonate. PLoS One. 2011;6(8):e23038. doi: 10.1371/journal.pone.0023038. Epub 2011 Aug 4. PMID: 21829686; PMCID: PMC3150387.

Maccarrone M, Barboni B, Paradisi A, Bernabò N, Gasperi V, Pistilli MG, Fezza F, Lucidi P, Mattioli M. Characterization of the endocannabinoid system in boar spermatozoa and implications for sperm capacitation and acrosome reaction. J Cell Sci. 2005 Oct 1;118(Pt 19):4393-404. doi: 10.1242/jcs.02536. Epub 2005 Sep 6. PMID: 16144868.

Bernabò N, Valbonetti L, Greco L, Capacchietti G, Ramal Sanchez M, Palestini P, Botto L, Mattioli M, Barboni B. Aminopurvalanol A, a Potent, Selective, and Cell Permeable Inhibitor of Cyclins/Cdk Complexes, Causes the Reduction of <i>in Vitro</i> Fertilizing Ability of Boar Spermatozoa, by Negatively Affecting the Capacitation-Dependent Actin Polymerization. Front Physiol. 2017 Dec 22;8:1097. doi: 10.3389/fphys.2017.01097. PMID: 29312003; PMCID: PMC5744433.

Oddi S, Bernabò N, Di Tommaso M, Angelucci CB, Bisicchia E, Mattioli M, Maccarrone M. DNA uptake in swine sperm: effect of plasmid topology and methyl- beta-cyclodextrin-mediated cholesterol depletion. Mol Reprod Dev. 2012 Dec;79(12):853-60. doi: 10.1002/mrd.22124. Epub 2012 Oct 30. PMID: 23071005.

Bernabò N, Pistilli MG, Mattioli M, Barboni B. Role of TRPV1 channels in boar spermatozoa acquisition of fertilizing ability. Mol Cell Endocrinol. 2010 Jul 29;323(2):224-31. doi: 10.1016/j.mce.2010.02.025. Epub 2010 Feb 26. PMID: 20219627.

Bernabò N, Berardinelli P, Mauro A, Russo V, Lucidi P, Mattioli M, Barboni B. The role of actin in capacitation-related signaling: an in silico and in vitro study. BMC Syst Biol. 2011 Mar 30;5:47. doi: 10.1186/1752-0509-5-47. PMID:21450097; PMCID: PMC3079638.

García-Martínez S, Latorre R, Sánchez-Hurtado MA, Sánchez-Margallo FM, Bernabò N, Romar R, López-Albors O, Coy P. Mimicking the temperature gradient between the sow's oviduct and uterus improves in vitro embryo culture output. Mol Hum Reprod. 2020 Oct 1;26(10):748-759. doi: 10.1093/molehr/gaaa053. PMID:32647896.

Bernabò N, Machado-Simoes J, Valbonetti L, Ramal-Sanchez M, Capacchietti G,Fontana A, Zappacosta R, Palestini P, Botto L, Marchisio M, Lanuti P, Ciulla M, Di Stefano A, Fioroni E, Spina M, Barboni B. Graphene Oxide increases mammalian spermatozoa fertilizing ability by extracting cholesterol from their membranes and promoting capacitation. Sci Rep. 2019 May 31;9(1):8155. doi:10.1038/s41598-019-44702-5. PMID: 31148593; PMCID: PMC6544623.

Coherently with these data we set up the duration of spermatozoa exposure to the materials 1.5h, i.e., the time required to reach the plateau level in percentage of capacitating spermatozoa.

About the lack of use of FBS, we adopted this approach for a very specific reason:

As it is known it contains several classes of proteins and other macromolecules. They bind with different mechanisms either the materials or the surface of spermatozoa, with two relevant effects:

  • They avoid the direct surfaces interaction between materials and cells.
  • They could promote important changes in cell membrane function (ex. Extracting cholesterol)

Bothe these effects are undesirable interferences in the experimental system (in our set-up) so we avoided the presence of FBS. Actually, to manage this problem through the years, we developed a system where it is possible to induce full capacitation of boar spermatozoa without external proteins. See for references:

Maccarrone, M., Barboni, B., Paradisi, A., Bernabò, N.Nicola, Gasperi, V., Pistilli, M.G., Fezza, F., Lucidi, P., Mattioli, M. Characterization of the endocannabinoid system in boar spermatozoa and implications for sperm capacitation and acrosome reaction (2005) Journal of Cell Science, 118 (19), pp. 4393-4404.

Bernabò, N., Palestini, P., Botto, L., Pistilli, M.G., Falasca, G., Gloria, A., Mattioli, M., Barboni, B. Lipidic microdomain reorganization during the in vitro capacitation of boar spermatozoa

(2009) Veterinary Research Communications, 33 (SUPPL. 1), pp. S81-S83.

Bernabò, N., Pistilli, M.G., Mattioli, M., Barboni, B. Role of TRPV1 channels in boar spermatozoa acquisition of fertilizing ability (2010) Molecular and Cellular Endocrinology, 323 (2), pp. 224-231.

Botto, L., Bernabò, N., Palestini, P., Barboni, B. Bicarbonate induces membrane reorganization and CBR1 and TRPV1 endocannabinoid receptor migration in lipid microdomains in capacitating boar spermatozoa (2010) Journal of Membrane Biology, 238 (1-3), pp. 33-41.

Bernabò, N., Berardinelli, P., Mauro, A., Russo, V., Lucidi, P., Mattioli, M., Barboni, B.The role of actin in capacitation-related signaling: An in silico and in vitro study

(2011) BMC Systems Biology, 5, art. no. 47, .

Barboni, B., Bernabò, N., Palestini, P., Botto, L., Pistilli, M.G., Charini, M., Tettamanti, E., Battista, N., Maccarrone, M., Mattioli, M. Type-1 Cannabinoid receptors reduce membrane fluidity of capacitated boar sperm by impairing their activation by bicarbonate (2011) PLoS ONE, 6 (8), art. no. e23038, .

Bernabò, N., Valbonetti, L., Greco, L., Capacchietti, G., Sanchez, M.R., Palestini, P., Botto, L., Mattioli, M., Barboni, B. Aminopurvalanol A, a potent, selective, and cell permeable inhibitor of Cyclins/Cdk complexes, causes the reduction of in vitro fertilizing ability of boar spermatozoa, by negatively affecting the capacitation-dependent actin polymerization  (2017) Frontiers in Physiology, 8 (DEC), art. no. 1097, 

Bernabò, N., Fontana, A., Sanchez, M.R., Valbonetti, L., Capacchietti, G., Zappacosta, R., Greco, L., Marchisio, M., Lanuti, P., Ercolino, E., Barboni, B. Graphene oxide affects in vitro fertilization outcome by interacting with sperm membrane in an animal model (2018) Carbon, 129, pp. 428-437.

Bernabò, N., Machado-Simoes, J., Valbonetti, L., Ramal-Sanchez, M., Capacchietti, G., Fontana, A., Zappacosta, R., Palestini, P., Botto, L., Marchisio, M., Lanuti, P., Ciulla, M., Di Stefano, A., Fioroni, E., Spina, M., Barboni, B. Graphene Oxide increases mammalian spermatozoa fertilizing ability by extracting cholesterol from their membranes and promoting capacitation (2019) Scientific Reports, 9 (1), art. no. 8155, .

I have some minor points that should be answered:

-CT because its antioxidant properties could have a benefit for sperm, but why analyze MoS2? The authors should improve the argument for which they have analyzed this.

Reply: Thanks for your question. Two-dimensional graphene-like molybdenum disulfide is a new material that has several unique properties which allow it potential applications. To our knowledge, and thanks to the numerous experiments carried out by our group on graphene oxide (GO) and spermatozoa, we considered MoS2 to be an excellent candidate for this study in order to propose this interesting material in the biological field. Noteworthy, we introduced an innovative use for catechins to exfoliate MOS2. Indeed, balancing oxidative stress is one of the main aspects of the capacitation process.

-line 157: Depend of the animal species that “spermatozoa that have lost their acrosome are unable to fertilize an egg”, in mice it has been demonstrated that sperm without acrosome can fertilize. Qualify the phrase.

Reply: Thank you for your suggestion; we have modified the sentence (Please see L160).

-Fig 2: In material and methods the authors indicate that they analyse the status of the acrosome of living and dead, indicate what figure 2 corresponds to. Indicate if the biological replicas are from three different individuals or from three ejaculates (same for the rest of figures).

Reply: Thank you for your comment. The biological replicates are from different boars. We added the information. L172

- Presenting 35% of fertilized oocytes is a very low figure, current IVF systems in pig hardly reach 50% to 60% efficiency, similar to that obtained with CT in this manuscript.

Reply: We set up our experimental system to be in an intermediate level of performance. We avoid reaching the maximum percentage of fertilized oocytes in control condition to be able to detect potential positive effect of the materials we tested. For this reason, we intentionally set the performance of the system below the maximum and over the minimum, thus we are able to identify both negative and positive effects of treatments.

-Indicate the number of fertilized oocytes per group in the four independent experiments that indicate that they have performed.

Reply: Thank you for your suggestion; we have added the number of fertilized oocytes per group (Please see L471).

-line 278, the authors say that their “findings support the evidence from previous studies showing that the addition 278 of CT to the extender had a positive effect on sperm motility of caprine sperm”.  In previous manuscripts the authors have reported 90% of fertilized oocytes (reference 63 used in M&M for indicated the protocol used in this work for IVF).

Reply: 1) we indicated the reference 63 (Maccarrone et al. Characterization of the endocannabinoid system in boar spermatozoa and implications for sperm ca-pacitation and acrosome reaction. J. Cell Sci. 2005, 118, 4393–4404, doi:10.1242/jcs.02536.) as reference for the sperm washing protocol: in detail we used sperm doses purchased form a market supplier (as described in M&M) and we removed the extender and the residual seminal plasma by centrifugation through a two-step discontinuous gradient of 35% and 70% isotonic Percoll. After removal of the supernatant layers, the resultant loose pellet was resuspended in residual 70% Percoll and washed by centrifugation at 800 g for 10 minutes in Dulbecco’s medium with Ca2+ and Mg2+. BSA was omitted because it could interfere with the direct interaction of sperm surface and materials.

  • In that paper we reported a percentage of around 50% of spermatozoa undergoing acrosome reaction after the co-incubation (1/1 vol/vol) with solubilized zona pellucida, which is a completely different way to assess the sperm capacitation status.

-Why has motility not been analysed in this work? Why has viability not been analysed if CT is related to oxidative stress? To analyse the toxicity of these components it is necessary to analyse motility and viability.

Reply: Viability was inferred by the other biochemical parameters examined. About motility we avoid its assessment for two reasons: 1) it is not obviously related with fertilizing ability of spermatozoa (while IVF is); 2) the absence of protein in the medium and the presence of materials flocks promote the aggregation of spermatozoa and their adherence to different surfaces, thus interfering with the CASA system, making unreliable the evaluation of motility.

Reviewer 4 Report

The manuscript presents data about addition of catechin and catechin-functionalized MoS2 nanoflakes to swine sperm capacitation medium. Nowadays there is an increasing trend of studying nature-derived, chemically defined substances and biomaterials as additives to sperm media and therefore using catechins and MoS2 Nanoflakes in this study presents modern approach. The authors evaluated many parameters related to sperm capacitation using various methods, which provided wide and comprehensive insight into the studied subject. In vitro fertilization has been performed as a last step. The experiment was well designed. The manuscript presents high scientific value, however, I have a feeling that in many points it is unclear and some important information are missing. I have also doubts about the interpretation of the in-vitro results.

First of all, it is not explained in the introduction why the authors have chosen MoS2 for their research – being ‘interesting candidate’ (Line 44) does not seem sufficient. Why authors thought it would be better than graphene oxide? Especially because the author mentioned potential toxicity (Lines 56-59) without giving any details – this aspect should be extended.

Secondly, the aim is not defined outright.

Thirdly, it is not clear how many samples were analysed (n=?) for each parameter and semen from how many boars was used? This should be added to text and figures. Additionally, information about animals (number, age, housing, feeding, frequency of semen collection etc) is missing in the Material and Methods part and there is no Ethical Statement included.

Fourthly, presentation of flow cytometry results is strange and very misleading. Generally, figures and tables should stand alone – the reader should be able to understand them without reading the whole article, which was impossible here. For the average reader words ‘frequency’ and ‘intervals’ are strongly associated with waves and time, making graphs hard to understand. ‘Percentage of spermatozoa’ and ‘fluorescence intensity’ would be much better. The units (a.u) should be explained here as well. In the description is should be stated, what low/high fluorescence intensity means for each staining.

Besides, why such form of data analysis? Typically, positive and negatives flow cytometry controls allows to set thresholds for ‘high’ and ‘low’ (sometimes ‘medium’ as well) values of examined parameter and then each spermatozoon is assigned to given subpopulation. Reduction of groups into 2-3 (instead of  41-101) makes interpretation of the results easier (both for the researcher and the reader). Moreover, it was written that viability staining was performed simultaneously, but there are no information about it in the results. Finally, Figures 3, 4 and 5 are of poor quality.

Last but not least (and in fact most important), why Authors regard polyspermy as an evidence of good fertilizing ability? Most of literature data agree that polyspermy is a huge problem during in-vitro fertilization in pigs and a lot of research has been performed to decrease its occurrence. Was the number of polyspermic oocytes included in total penetrated? If so, I wouldn’t say the results of in vitro fertilization were better – in fact, they could have been worse, if less oocytes were fertilized properly (by one spermatozoon). This may change the conclusions completly. 

Additional question to this – was the penetration assessed under stereomicroscope or after fluorescent staining?

To sum up, despite properly performed experiment and high scientific value, abovementioned issues require major revision and/or answers. There are also some minor mistakes (see below).

Minor remarks:

Lines 69-72 ‘In the present study, nanoflakes of MoS2 functionalized with catechins and catechins were used to study their impact on swine spermatozoa functional parameters during capacitation. The complexes MoS2/CT and CT were used here to evaluate the effects on an important biological event as the in vitro capacitation.’ Second phrase duplicates the first one - to be corrected

Results and Discussion are have been combined into one section, which does not follow Journal’s ‘Instruction for Authors’ (https://www.mdpi.com/journal/ijms/instructions). Also, such form does not allow readers to quickly study the results. 

Part 2.1. ‘Preparation and characterization of water-phase exfoliated MoS2/CT nanoflakes’ was not the scope of this study and it has been already published, therefore it should not be included as results here. Also, it can be discusses less extensively.

Lines 332 and further for centrifugation: x g

Figure 8: please add capacitation time to the picture and description

Line 369-370: was motility estimated before capacitation or after?

Line 417 and further: pKa à PKA

Lines 462 and 472 state different number of in-vitro experiments.

Line 180, 487 and description of graphs in Supp. Material – add time unit [h] to 0 and 1.5. Do the comparison of CTRL at 0 and 1,5h prove that capacitation occurred? Are there any statistical differences? If so, they should be marked on the graphs.

Author Response

The manuscript presents data about addition of catechin and catechin-functionalized MoS2 nanoflakes to swine sperm capacitation medium. Nowadays there is an increasing trend of studying nature-derived, chemically defined substances and biomaterials as additives to sperm media and therefore using catechins and MoS2 Nanoflakes in this study presents modern approach. The authors evaluated many parameters related to sperm capacitation using various methods, which provided wide and comprehensive insight into the studied subject. In vitro fertilization has been performed as a last step. The experiment was well designed. The manuscript presents high scientific value, however, I have a feeling that in many points it is unclear and some important information are missing. I have also doubts about the interpretation of the in-vitro results.

First of all, it is not explained in the introduction why the authors have chosen MoS2 for their research – being ‘interesting candidate’ (Line 44) does not seem sufficient. Why authors thought it would be better than graphene oxide? Especially because the author mentioned potential toxicity (Lines 56-59) without giving any details – this aspect should be extended.

Reply: First of all, we would like to thank Rev 3 for his/her valuable comments and opinions, which we have taken into account to improve our manuscript. Two-dimensional graphene-like molybdenum disulfide is a new material that has several unique properties which allow it potential applications. To our knowledge, and thanks to the numerous experiments carried out by our group on graphene oxide (GO) and spermatozoa, we considered MoS2 to be an excellent candidate for this study in order to propose this interesting material in the biological field. Noteworthy, we introduced an innovative use for catechins to exfoliate mos2. Indeed, balancing oxidative stress is one of the main aspects of the capacitation process. Potential toxicity was mentioned in the text; however, to our knowledge the toxicity of this complex has not been demonstrated yet in spermatozoa. At the concentrations used in this study, we found no toxic effects.

Secondly, the aim is not defined outright.

Reply: The aim of our work is presented in the Introduction section, L73-85.

Thirdly, it is not clear how many samples were analysed (n=?) for each parameter and semen from how many boars was used? This should be added to text and figures. Additionally, information about animals (number, age, housing, feeding, frequency of semen collection etc) is missing in the Material and Methods part and there is no Ethical Statement included.

Reply: We agree. Accordingly, the reviewers’ suggestion we added the number of experiments. No Ethical statement is needed for this study, since we used commercially available doses of semen, as stated in the materials and methods section, L367 and 449.

Fourthly, presentation of flow cytometry results is strange and very misleading. Generally, figures and tables should stand alone – the reader should be able to understand them without reading the whole article, which was impossible here. For the average reader words ‘frequency’ and ‘intervals’ are strongly associated with waves and time, making graphs hard to understand. ‘Percentage of spermatozoa’ and ‘fluorescence intensity’ would be much better. The units (a.u) should be explained here as well. In the description is should be stated, what low/high fluorescence intensity means for each staining. Besides, why such form of data analysis? Typically, positive and negatives flow cytometry controls allows to set thresholds for ‘high’ and ‘low’ (sometimes ‘medium’ as well) values of examined parameter and then each spermatozoon is assigned to given subpopulation. Reduction of groups into 2-3 (instead of  41-101) makes interpretation of the results easier (both for the researcher and the reader). Moreover, it was written that viability staining was performed simultaneously, but there are no information about it in the results. Finally, Figures 3, 4 and 5 are of poor quality.

Reply: Thank you for the comments and suggestion. We understand that this kind of analysis could be complex and difficult to comprehend at a first approach. In this study, we followed a complex flow cytometry analysis in an effort to consider and analyse every single spermatozoon and not the fluorescence means of a whole population (or subpopulation). The reason of this strategy is mainly due to the great interest of the scientific community in finding the features of the fertilizing spermatozoon, the one that fertilize, since it is always a single (or a couple) spermatozoa to fertilize an egg and not a whole population. Thus, we consider the fluorescence of each spermatozoa, distributing them by intervals of fluorescence (i.e., in reduced groups) that allowed to infer the trends in the events studied (intracellular calcium concentration, mitochondrial activity, membrane disorder) to take the maximum from our experiments and from our results. In this case, we can affirm that there are no differences between the groups for the parameters studied.

Last but not least (and in fact most important), why Authors regard polyspermy as an evidence of good fertilizing ability? Most of literature data agree that polyspermy is a huge problem during in-vitro fertilization in pigs and a lot of research has been performed to decrease its occurrence. Was the number of polyspermic oocytes included in total penetrated? If so, I wouldn’t say the results of in vitro fertilization were better – in fact, they could have been worse, if less oocytes were fertilized properly (by one spermatozoon). This may change the conclusions completely. 

Reply: Thank you for your comment. Here, we used the IVF as a functional test for spermatozoa, without the aim of obtaining embryos. It means that our work was set up to evaluate the capacitation status of sperm cells in a specific context: after exposure during capacitation to MoS2 functionalized with catechins and catechins. As it has been reported in literature that in swine, in vitro maturated oocytes are partially unable to avoid the polyspermic fertilization, thus in specific conditions in term of sperm/oocytes ratio and length of coincubation, it is possible to obtain polyspermic oocytes as a marker of the fertilizing ability.

Additional question to this – was the penetration assessed under stereomicroscope or after fluorescent staining?

Reply: Thank you for your question; the penetration rate was evaluated after staining with Hoechst 33342 and assessed under the fluorescence microscope.

To sum up, despite properly performed experiment and high scientific value, above mentioned issues require major revision and/or answers. There are also some minor mistakes (see below).

Minor remarks:

Lines 69-72 ‘In the present study, nanoflakes of MoS2 functionalized with catechins and catechins were used to study their impact on swine spermatozoa functional parameters during capacitation. The complexes MoS2/CT and CT were used here to evaluate the effects on an important biological event as the in vitro capacitation.’ Second phrase duplicates the first one - to be corrected.

Reply: We apologize for the mistake; we have removed the second sentence.

Results and Discussion are have been combined into one section, which does not follow Journal’s ‘Instruction for Authors’ (https://www.mdpi.com/journal/ijms/instructions). Also, such form does not allow readers to quickly study the results. 

Reply: Thank you for your proposal. Although we consider your comment very valuable, and we have agreed with you in other manuscripts, we believe that, in this specific case, results and discussions should be merged, since it promotes the work done and makes understanding of the results more immediate, following a fluid timeline. We apologize for the inconveniences.

Part 2.1. ‘Preparation and characterization of water-phase exfoliated MoS2/CT nanoflakes’ was not the scope of this study and it has been already published, therefore it should not be included as results here. Also, it can be discusses less extensively.

Reply: Thank your for your comment. Although different materials and compounds have been previously prepared in a similar manner, this is the first time that MoS2 has been prepared and exfoliated with catechins. Moreover, we consider that the characterization of the compound is of great importance to evidence in detail what the spermatozoa are cultured/treated with. The material here presented could not be considered a commercial material, dispersed in a conventional way; thus, the characterization is necessary as a matter of transparency and reproducibility.

Lines 332 and further for centrifugation: x g

Reply: We apologize for the mistakes; we have corrected the sentences.

Figure 8: please add capacitation time to the picture and description.

Reply: In keeping with the reviewer's suggestion, we have added capacitation time to the picture and description.

Line 369-370: was motility estimated before capacitation or after?

Reply: Sperm motility was estimated before the capacitation and only samples with sperm motility > 90% were considered for further analyses. This information was added (L374)

Line 417 and further: pKa à PKA

Reply: We apologize for the mistake, we have corrected the sentence (Please see L422)

Lines 462 and 472 state different number of in-vitro experiments.

Reply: We apologize for the mistake; the number of IVF experiment is four. We have corrected the sentence (Please see L477)

Line 180, 487 and description of graphs in Supp. Material – add time unit [h] to 0 and 1.5. Do the comparison of CTRL at 0 and 1,5h prove that capacitation occurred? Are there any statistical differences? If so, they should be marked on the graphs.

Reply: Thank you for your very interesting question. In this study, we followed a complex flow cytometry analysis in an effort to consider and analyse every single spermatozoon and not the fluorescence means of a whole population (or subpopulation). The reason of this strategy is mainly due to the great interest of the scientific community in finding the features of the fertilizing spermatozoon, the one that fertilize, since it is always a single (or a couple) spermatozoon to fertilize an egg and not a whole population. Thus, we consider the fluorescence of each spermatozoa, distributing them by intervals of fluorescence (i.e., in reduced groups) that allowed to infer the trends in the events studied (intracellular calcium concentration, mitochondrial activity, membrane disorder) to take the maximum from our experiments and from our results. In this case, we can affirm that there are no differences between the groups for the parameters studied. However, as you noted, differences are evidenced when comparing the CTRL at T0 and T1.5 h of capacitation for the three events studied, as can be observed in the Supplementary Materials, confirming that the spermatozoa at T1.5 follow the classical trends of fertility ability acquisition. We added time unit [h] to Supplementary Material 1.

Round 2

Reviewer 1 Report

Catechin versus MoS2 nanoflakes functionalized with catechin: 2 improving the sperm fertilizing ability – an in vitro study on 3 swine model.

Second Review

Thanks to Costanza Cimini et al. for this study. Authors have addressed some questions but some remained un answered.

Q1. Regarding question 4 in first round of review, your answer is “In this study, we did not evaluate single bands, but the activity of an important protein such as PKA, as well as a pattern of tyrosine phosphorylation. For this reason, we did not crop the membranes. All the antibodies used are specified within the Materials and Methods section. Moreover, the scope of this kind of analysis is not to “quantify” (considering the limits of WB technique, since it represents a semi-quantitative analysis) the protein expression, but to confirm the absence of differences in the activity of a protein and in the phosphorylation patterns of a protein cocktail.

Authors have mentioned some references to defense their findings. After going through references I would suggest to mark the ladder molecular weight on left hand side.

Q2. Regarding question 6 in first round of review, your answer is “The physiology of calcium control is very complex in spermatozoa either in resting conditions as well as during capacitation and acrosome reaction. Indeed, Ca2+ intracellular concentration is one of the key modulators of sperm function and the most important signaling system, involved in virtually all the biochemical processes involved in capacitation.

It involves thousands of different kinds of channels and pumps, see for references:

I agree that there are different kinds of channels and pumps for catsper but I would not agree with this reply, For this reason, we concentrated our attention on the final event (the variation in Calcium concentration) instead that on the involved molecules, that are a myriad.” One research cannot get all the required results. Therefore, may be in future your work would be referred with some close related results. However, this is just excuse to ignore.

Please find article published related with capacitation and catsper in below link.

https://www.mdpi.com/1422-0067/23/23/14646

https://elifesciences.org/articles/62043

Author Response

Thanks to Costanza Cimini et al. for this study. Authors have addressed some questions but some remained un answered.

Q1. Regarding question 4 in first round of review, your answer is “In this study, we did not evaluate single bands, but the activity of an important protein such as PKA, as well as a pattern of tyrosine phosphorylation. For this reason, we did not crop the membranes. All the antibodies used are specified within the Materials and Methods section. Moreover, the scope of this kind of analysis is not to “quantify” (considering the limits of WB technique, since it represents a semi-quantitative analysis) the protein expression, but to confirm the absence of differences in the activity of a protein and in the phosphorylation patterns of a protein cocktail.”

Authors have mentioned some references to defense their findings. After going through references I would suggest to mark the ladder molecular weight on left hand side.

Reply: We would like to thank Rev. 1 for his/her time in reviewing again our manuscript, and for the interesting comments and suggestions. We have addressed your request and added the molecular weight on left hand side.

Q2. Regarding question 6 in first round of review, your answer is “The physiology of calcium control is very complex in spermatozoa either in resting conditions as well as during capacitation and acrosome reaction. Indeed, Ca2+ intracellular concentration is one of the key modulators of sperm function and the most important signaling system, involved in virtually all the biochemical processes involved in capacitation.”

It involves thousands of different kinds of channels and pumps, see for references:

I agree that there are different kinds of channels and pumps for catsper but I would not agree with this reply, “For this reason, we concentrated our attention on the final event (the variation in Calcium concentration) instead that on the involved molecules, that are a myriad.” One research cannot get all the required results. Therefore, may be in future your work would be referred with some close related results. However, this is just excuse to ignore.

Please find article published related with capacitation and catsper in below link.

https://www.mdpi.com/1422-0067/23/23/14646

https://elifesciences.org/articles/62043

Reply: We agree with Rev. 1 regarding this issue; however, the study of Catsper channels was not the scope of the present study. Since we agree with the Rev. 1 in the importance of this channels, we have added some lines and the suggested references to our manuscript (L207-208).

Reviewer 3 Report

With the changes introduced, the manuscript has improved considerably and is ready to publish.

Author Response

Thank you.

Reviewer 4 Report

Dear Authors,

Most of the comments were not addressed satisfactory to me. However, I am fully aware of the fact that my remarks were mostly about the style, not about the scientific matter. Many things are disputable (e.g. ‘single-sperm’ data analysis or interpretation of polyspermy) and both sides have arguments supporting their opinion. I evaluate scientific content as very high. But at the same time I look on the article from the reader side – and I find it not easy to follow.

What I insist must be improved:

-       Clearly state ‘n’ number

-       Information about preparation and characterization of water-phase exfoliated MoS2/CT nanoflakes – the authors wrote in the original manuscript that it has been published previously.

 Please see detailed comments in the attached file.

Author Response

Dear Authors,

Most of the comments were not addressed satisfactory to me. However, I am fully aware of the fact that my remarks were mostly about the style, not about the scientific matter. Many things are disputable (e.g. ‘single-sperm’ data analysis or interpretation of polyspermy) and both sides have arguments supporting their opinion. I evaluate scientific content as very high. But at the same time I look on the article from the reader’s side – and I find it not easy to follow.

What I insist must be improved:

  • Clearly state ‘n’ number
  • Information about preparation and characterization of water-phase exfoliated MoS2/CT nanoflakes – the Authors wrote in the manuscript that it has been published previously.

Please see below detailed comments in red.

The manuscript presents data about addition of catechin and catechin-functionalized MoS2 nanoflakes to swine sperm capacitation medium. Nowadays there is an increasing trend of studying nature-derived, chemically defined substances and biomaterials as additives to sperm media and therefore using catechins and MoS2 Nanoflakes in this study presents modern approach. The authors evaluated many parameters related to sperm capacitation using various methods, which provided wide and comprehensive insight into the studied subject. In vitro fertilization has been performed as a last step. The experiment was well designed. The manuscript presents high scientific value, however, I have a feeling that in many points it is unclear and some important information are missing. I have also doubts about the interpretation of the in-vitro results.

First of all, it is not explained in the introduction why the authors have chosen MoS2 for their research – being ‘interesting candidate’ (Line 44) does not seem sufficient. Why authors thought it would be better than graphene oxide? Especially because the author mentioned potential toxicity (Lines 56-59) without giving any details – this aspect should be extended.

Reply: First of all, we would like to thank Rev 3 for his/her valuable comments and opinions, which we have taken into account to improve our manuscript. Two-dimensional graphene-like molybdenum disulfide is a new material that has several unique properties which allow it potential applications. To our knowledge, and thanks to the numerous experiments carried out by our group on graphene oxide (GO) and spermatozoa, we considered MoS2 to be an excellent candidate for this study in order to propose this interesting material in the biological field. Noteworthy, we introduced an innovative use for catechins to exfoliate mos2. Indeed, balancing oxidative stress is one of the main aspects of the capacitation process. Potential toxicity was mentioned in the text; however, to our knowledge the toxicity of this complex has not been demonstrated yet in spermatozoa. At the concentrations used in this study, we found no toxic effects.

My question is still not answered – why the Authors thought it would be better than graphene oxide? Why the Authors gave up with GO and changed to MoS2?

Reply: we decided to use other material than GO because we think that they could provide interesting information. We are still working on the application of GO for sperm selection and membrane engineering – we hope to present our new results soon, but they are out of the focus of the present work – and the use of MoS2 and catechins is part of the same approach. In particular we are exploring the possible application of those materials for technological purposes. Since GO seems to increase the sperm capacitation by interacting with membranes (in particular by promoting the cholesterol extraction) we guess if that effect could be dependent on surface chemistry or/and on geometry of the system. That’s why here we used a material with a different surface chemistry and an almost similar architecture. The acquisition of data will gradually lead us to develop hypothesis and models in that regard.

Secondly, the aim is not defined outright.

Reply: The aim of our work is presented in the Introduction section, L73-85.

Nothing changed. By clearly defined aim I understand one, concise phrase, not a whole paragraph.

Reply: Thank you for the suggestion; we have reassumed the aim in a single sentence, as requested (L73-74).

Thirdly, it is not clear how many samples were analysed (n=?) for each parameter and semen from how many boars was used? This should be added to text and figures. Additionally, information about animals (number, age, housing, feeding, frequency of semen collection etc) is missing in the Material and Methods part and there is no Ethical Statement included.

Reply: We agree. Accordingly, the reviewers’ suggestion we added the number of experiments. No Ethical statement is needed for this study, since we used commercially available doses of semen, as stated in the materials and methods section, L367 and 449.

Number of experiments is not the same as number of samples. Information ‘(from different boars)‘ was added, without giving the number of animals, my remark was not answered correctly. Please clearly state the ‘N’ number. (i.e. if semen from three boars was used and the experiment was repeated three times for each boar, then n=9).

Replay: We apologize for the mistake, we added the missing information, please see Line 188-194-200

Fourthly, presentation of flow cytometry results is strange and very misleading. Generally, figures and tables should stand alone – the reader should be able to understand them without reading the whole article, which was impossible here. For the average reader words ‘frequency’ and ‘intervals’ are strongly associated with waves and time, making graphs hard to understand. ‘Percentage of spermatozoa’ and ‘fluorescence intensity’ would be much better. The units (a.u) should be explained here as well. In the description is should be stated, what low/high fluorescence intensity means for each staining. Besides, why such form of data analysis? Typically, positive and negatives flow cytometry controls allows to set thresholds for ‘high’ and ‘low’ (sometimes ‘medium’ as well) values of examined parameter and then each spermatozoon is assigned to given subpopulation. Reduction of groups into 2-3 (instead of  41-101) makes interpretation of the results easier (both for the researcher and the reader). Moreover, it was written that viability staining was performed simultaneously, but there are no information about it in the results. Finally, Figures 3, 4 and 5 are of poor quality.

Reply: Thank you for the comments and suggestion. We understand that this kind of analysis could be complex and difficult to comprehend at a first approach. In this study, we followed a complex flow cytometry analysis in an effort to consider and analyse every single spermatozoon and not the fluorescence means of a whole population (or subpopulation). The reason of this strategy is mainly due to the great interest of the scientific community in finding the features of the fertilizing spermatozoon, the one that fertilize, since it is always a single (or a couple) spermatozoa to fertilize an egg and not a whole population. Thus, we consider the fluorescence of each spermatozoa, distributing them by intervals of fluorescence (i.e., in reduced groups) that allowed to infer the trends in the events studied (intracellular calcium concentration, mitochondrial activity, membrane disorder) to take the maximum from our experiments and from our results. In this case, we can affirm that there are no differences between the groups for the parameters studied.

I do not agree that such individual sperm approach is superior to subpopulations. How do you know fluorescence intensity for this one sperm cell which fertilize the oocyte? How do you know that a spermatozoon with fluorescence intensity 18000 a.u. has better fertilizing potential than one with 20000 a.u.? That would make sense only if there is a strong correlation between fluorescence intensity and fertilizing ability (is this reported for studied parameters?).

Can you provide any publication showing data this way?

However, I can accept that the authors would like to stay with this presentation of data – it does not change the content, only feasibility to the reader.

Quality of figures did not change. Question about viability was not answered as well – what is the point of dual staining, if the second fluorochrome is not taken into consideration?

Reply: Thank you for your comments and the open discussion, which we consider very interesting. Actually, we did not say that a spermatozoon with a higher fluorescence intensity is the one able to fertilize the oocyte, but that the possibility to distinguish several subpopulations among the whole population opens new horizons for the selection and the future study of the single populations, to evaluate their single fertilizing potential. For this reason, we did not focus our study in evaluating the significance among the interval groups, since we could not affirm that these differences are the consequence of a higher or lower fertilizing potential.

This kind of analysis has been already published by our group, in collaboration we other colleagues and some experts on flow cytometry. Please see this reference:

https://onlinelibrary.wiley.com/doi/full/10.1111/andr.12971

Thus, in this specific case, we searched for the differences between the studied groups, not about the differences in one single group (thus, from a functional point of view). We chose this analysis to confirm the results obtained in terms of percentages that often can carry researches to miss important information. In any case, we are open for discussion.

Regarding the viability, we confirm that the cells used for the analysis are alive, thus previously selected as positive or negative for the viability fluorochrome. This was already stated in the materials and methods section, L416-42

“To that, the columns “FCS”, “SSC”, “Fluo-4AM”, “PI”, “DilC12”, “Mito” and “NIR” with the data from the 100000 events acquired were selected and filtered following these criteria: forward scatter (FCS) between 35000 and 135000 arbitrary units (a.u); side scatter (SSC) between 20000 and 145000 a.u; Fluo 3-AM, M540 and Mitotracker Red >0 a.u; PI between 0 and 30000 a.u; and Near Infrared between 0 and 20000 a.u. Then, data were treated and subdivided in intervals of fluorescence intensity as follow: 41 intervals for intracellular calcium (from 0 to 20000 a.u, 500 a.u range); 101 intervals for membrane disorder (from 0 to 50000 AU, 500 a.u range); 61 intervals for mitochondrial activity (from 0 to 30000 a.u, 500 a.u range). Above the 98% of the data fitted within this range.”

Last but not least (and in fact most important), why Authors regard polyspermy as an evidence of good fertilizing ability? Most of literature data agree that polyspermy is a huge problem during in-vitro fertilization in pigs and a lot of research has been performed to decrease its occurrence. Was the number of polyspermic oocytes included in total penetrated? If so, I wouldn’t say the results of in vitro fertilization were better – in fact, they could have been worse, if less oocytes were fertilized properly (by one spermatozoon). This may change the conclusions completely. 

Reply: Thank you for your comment. Here, we used the IVF as a functional test for spermatozoa, without the aim of obtaining embryos. It means that our work was set up to evaluate the capacitation status of sperm cells in a specific context: after exposure during capacitation to MoS2 functionalized with catechins and catechins. As it has been reported in literature that in swine, in vitro maturated oocytes are partially unable to avoid the polyspermic fertilization, thus in specific conditions in term of sperm/oocytes ratio and length of coincubation, it is possible to obtain polyspermic oocytes as a marker of the fertilizing ability.

References given by the authors are from 1988 and 1997 (the second one citing the first one) and states: ‘ According to Hunter & Nichol (1988), the incidence and degree of polyspermy are an indication of number of capacitated spermatozoa’ (nothing about fertilizing ability, only capacitation marker). More recent publications report monospermic fertilization rates instead of polyspermy (e.g. https://doi.org/10.1016/j.theriogenology.2007.06.006, https://doi.org/10.5194/aab-64-265-2021) which I believe is more appropriate. If Authors insist on polyspermy, newer references should be added. Alternatively, sometimes in the literature polyspermy is used as ‘penetration ability’ marker. 

Reply: Actually, as stated by Rev. 3, here we adopted and experimental set-up to allow the polyspermic fertilzation, using that parameter as marker for spermatozoa penetartion ability, as we have done in other articles (Bernabò N et al. Graphene oxide affects in vitro fertilization outcome by interacting with sperm membrane in an animal model. Carbon Volume 129, April 2018, Pages 428-437; Cimini C et al. Pre-Treatment of Swine Oviductal Epithelial Cells with Progesterone Increases the Sperm Fertilizing Ability in an IVF Model. Animals (Basel). 2022 May 6;12(9):1191)

Additional question to this – was the penetration assessed under stereomicroscope or after fluorescent staining?

Reply: Thank you for your question; the penetration rate was evaluated after staining with Hoechst 33342 and assessed under the fluorescence microscope.

Information not added to the manuscript.

Reply: Sorry for the mistake, the information was added (please see Line 468-469)

Part 2.1. ‘Preparation and characterization of water-phase exfoliated MoS2/CT nanoflakes’ was not the scope of this study and it has been already published, therefore it should not be included as results here. Also, it can be discusses less extensively.

Reply: Thank your for your comment. Although different materials and compounds have been previously prepared in a similar manner, this is the first time that MoS2 has been prepared and exfoliated with catechins. Moreover, we consider that the characterization of the compound is of great importance to evidence in detail what the spermatozoa are cultured/treated with. The material here presented could not be considered a commercial material, dispersed in a conventional way; thus, the characterization is necessary as a matter of transparency and reproducibility.

The Authors themselves stated in the manuscript: ‘The catechin effectiveness in the exfoliation of MoS2 was proved in our previous work [12], where the exfoliation strategy was proposed, optimized, and the nanoflakes obtained were fully characterized.’ I do not have the access to referred publication, so I cannot check it. Results published once cannot be repeated in another publication – this is an important ethical issue.

 Reply: In this paper has been used the first-time this material (i.e., MoS2/CT) to perform studies on spermatozoa; the material characterization reported here is exclusive, and each Figure is not a repetition of previous papers. In the previous paper named, we proposed for the first time the synthesis (exfoliation strategy) of TMDs using different polyphenols, and we have optimized the same; no result of the other work is duplicated in this one. In the named previous work, we demonstrated the theory of how TMDs can be synthesized, but we didn't focus on MoS2 production with CT. On the other hand, in the present work, the materials synthesis optimization is not reported, also because it's off-topic. Precisely because we are aware of the total absence of ethical problems, we have referred to the previous work. 

Line 180, 487 and description of graphs in Supp. Material – add time unit [h] to 0 and 1.5. Do the comparison of CTRL at 0 and 1,5h prove that capacitation occurred? Are there any statistical differences? If so, they should be marked on the graphs.

Reply: Thank you for your very interesting question. In this study, we followed a complex flow cytometry analysis in an effort to consider and analyse every single spermatozoon and not the fluorescence means of a whole population (or subpopulation). The reason of this strategy is mainly due to the great interest of the scientific community in finding the features of the fertilizing spermatozoon, the one that fertilize, since it is always a single (or a couple) spermatozoon to fertilize an egg and not a whole population. Thus, we consider the fluorescence of each spermatozoa, distributing them by intervals of fluorescence (i.e., in reduced groups) that allowed to infer the trends in the events studied (intracellular calcium concentration, mitochondrial activity, membrane disorder) to take the maximum from our experiments and from our results. In this case, we can affirm that there are no differences between the groups for the parameters studied. However, as you noted, differences are evidenced when comparing the CTRL at T0 and T1.5 h of capacitation for the three events studied, as can be observed in the Supplementary Materials, confirming that the spermatozoa at T1.5 follow the classical trends of fertility ability acquisition. We added time unit [h] to Supplementary Material 1.

Do I understand correctly – Author’s ‘single-sperm’ approach does not allow to mark any significant differences on the graphs? I think it confirms that it is not the best way to analyze the results. If something is not statistically significant, then there are no differences – we have to assume that capacitation did not occur.

Looking at the main results , I can see differences in the curves for experimental groups – I think there would be statistical differences in ‘low’ and ‘high’ subpopulations for evaluated parameters.

Additionally, the presentation of the results in Supplementary materials is misleading – because of differences in the y-axis, on the plots for Ca2+ it looks like after 1,5h incubation the percentage of low-fluorescent cells is higher than at 0 point. The axes should have the same units range for 0h and 1,5h.

Reply: Thank you again for this second part of discussion on our flow cytometry analysis. As already explained above, evaluating the functional features of sperm was not the scope of this work. The scope of the current study was to evaluate, from different points of view (intracellular calcium concentration, mitochondrial activity and membrane fluidity modifications) the differences between the groups evaluated, i.e., the spermatozoa capacitated in the presence or absence of the non-physiological material, MoS2-catechins and catechins alone. Thus, we did not focused on the significance among the control groups at T0 and T1.5, we just added this information as a matter of 1) transparency and 2) supplementary information for the reader, in case of need. We apologize for the inconveniences, and we hope our answer could be considered as satisfactory now. In any case, as stated above, we remain ad your disposal for future discussions regarding this kind of analysis.